# A transcriptional constraint mechanism limits the homeostatic response to activity deprivation in mammalian neocortex

Vera Valakh, Derek Wise, Xiaoyue Aelita Zhu, Mingqi Sha, Jaidyn Fok, Stephen D Van Hooser, Robin Schectman, Isabel Cepeda, Ryan Kirk, Sean M O'Toole, Sacha B Nelson*

Department of Biology and Program in Neuroscience, Brandeis University, Waltham, United States

**Abstract** Healthy neuronal networks rely on homeostatic plasticity to maintain stable firing rates despite changing synaptic drive. These mechanisms, however, can themselves be destabilizing if activated inappropriately or excessively. For example, prolonged activity deprivation can lead to rebound hyperactivity and seizures. While many forms of homeostasis have been described, whether and how the magnitude of homeostatic plasticity is constrained remains unknown. Here, we uncover negative regulation of cortical network homeostasis by the PARbZIP family of transcription factors. In cortical slice cultures made from knockout mice lacking all three of these factors, the network response to prolonged activity withdrawal measured with calcium imaging is much stronger, while baseline activity is unchanged. Whole-cell recordings reveal an exaggerated increase in the frequency of miniature excitatory synaptic currents reflecting enhanced upregulation of recurrent excitatory synaptic transmission. Genetic analyses reveal that two of the factors, *Hlf* and *Tef*, are critical for constraining plasticity and for preventing life-threatening seizures. These data indicate that transcriptional activation is not only required for many forms of homeostatic plasticity but is also involved in restraint of the response to activity deprivation.

*For correspondence:
nelson@brandeis.edu

## Editor's evaluation

Homeostatic plasticity helps to maintain the stability of neural network activity. This study shows that activation of PAR bZIP family of transcription factors restrains homeostatic upregulation of network activity in response to activity deprivation in mouse brain slice cultures. The identification of an endogenous transcriptional program that limits upward homeostatic response and helps prevent aberrant activity associated with epilepsy and brain disorders is important, and the findings will be of interest to a broad neuroscience community.

## Introduction

Neuronal networks are equipped with a set of homeostatic plasticity mechanisms that enable them to rebalance activity following perturbations during development, learning, or disease. Homeostatic plasticity can alter the intrinsic excitability of individual neurons, as well as the strength and number of both excitatory and inhibitory synapses (*Davis, 2006*; *Turrigiano and Nelson, 2004*). Synaptic changes can occur post- (*Turrigiano et al., 1998*) or presynaptically (*Delvendahl and Müller, 2019*) and may scale quantal amplitudes and alter the frequency of quantal events. These changes are homeostatic

---

**eLife digest** The human brain is made up of billions of nerve cells called neurons which receive and send signals to one another. To avoid being over- or under-stimulated, neurons can adjust the strength of the inputs they receive by altering how connected they are to other nerve cells.

This process, known as homeostatic plasticity, is thought to be necessary for normal brain activity as it helps keep the outgoing signals of neurons relatively constant. However, homeostatic plasticity can lead to seizures if it becomes too strong and overcompensates for weak input signals. Regulating this process is therefore central to brain health, but scientists do not understand if or how it is controlled.

To address this, Valakh et al. analyzed the genes activated in neurons lacking incoming signals to find proteins that regulate homeostatic plasticity. This revealed a class of molecules called transcription factors (which switch genes on or off) that constrain the process. In brain samples from mice without these regulatory proteins, neurons received twice as much input, leading to an increase in brain activity resembling that observed during seizures. Valakh et al. confirmed this finding using live mice, which developed seizures in the absence of these transcription factors.

These findings suggest that this type of regulation to keep homeostatic plasticity from becoming too strong may be important. This could be especially vital as the brain develops, when the strength of connections between neurons changes rapidly. The discovery of the transcription factors involved provides a potential target for activating or restraining homeostatic plasticity. This control could help researchers better understand how the process stabilizes brain signaling.

---

because they occur in the direction needed to rebalance the network after activity perturbation. Especially during development, homeostatic mechanisms are strong and can be maladaptive if they overshoot or are activated inappropriately (*Nelson and Valakh, 2015*).

Despite recognition of the significance of homeostatic plasticity, whether and how its strength is normally constrained has remained unknown. Downregulation of the strength of homeostatic plasticity could potentially provide protection against inappropriate or excessive activation of these mechanisms. However, such negative regulators of homeostatic plasticity have not been previously described.

Here, we investigate the role of the proline and acidic amino acid-rich basic leucine zipper (PAR bZIP) family of transcription factors (TFs) in homeostatic plasticity. The family consists of hepatic leukemia factor (HLF), thyrotroph embryonic factor (TEF), and albumin D-site-binding protein (DBP) which are thought to act as transcriptional activators, as well as E4 Promoter-Binding Protein 4 (E4BP4, currently known as Nfil3) that acts as a transcriptional repressor (*Mitsui et al., 2001*). All four family members share the same DNA binding motif. DBP is a circadian gene controlled by CLOCK and oscillates with circadian rhythm (*Ripperger et al., 2000*). Other family members are also controlled by CLOCK (*Li et al., 2017*). However, while these TFs typically have oscillatory behavior in peripheral tissue, they do not oscillate in the brain outside of the SCN (*Gachon et al., 2004*). Loss of HLF, TEF, and DBP has been associated with epilepsy (*Gachon et al., 2004*; *Hawkins and Kearney, 2016*; *Rambousek et al., 2020*) suggesting they are important in regulating network activity.

In order to investigate the role that gene transcription plays in regulating the neuronal response to activity deprivation, we profiled changes in gene expression engaged during homeostatic plasticity and found that prolonged activity deprivation activates a robust transcriptional program involving the PARbZIP TF family members HLF and TEF. Both act to restrain the expression of homeostatic plasticity. While they have limited effects on network function at baseline, they strongly suppress the upregulation of homeostatic changes, mainly by regulating recurrent excitatory synaptic connections, although inhibition is also affected. Taken together, these results indicate that homeostatic plasticity is itself subject to activity-dependent regulation and is transcriptionally restrained by the PARbZIP TFs HLF and TEF.

# Results

## PARbZIP transcripts increase during activity deprivation

Certain forms of homeostatic plasticity require transcription (*Goold and Nicoll, 2010*; *Ibata et al., 2008*) and activity perturbation results in changes in gene expression (*Schaukowitch et al., 2017*). We hypothesized that among the differentially expressed genes, some induce or regulate homeostatic changes. To identify novel genes involved in the homeostatic response to activity deprivation, we measured gene expression in excitatory and inhibitory neurons in organotypic slice cultures following a global decrease in activity. We cut coronal slices including neocortex at P7 and cultured them for 5 days. During this culture period, the neurons reform connections, and spontaneous bouts of activity, termed upstates, emerge (*Johnson and Buonomano, 2007*; *Koch et al., 2010*). To broadly block activity, we applied the voltage-gated sodium channel blocker tetrodotoxin, TTX (0.5 µM), for

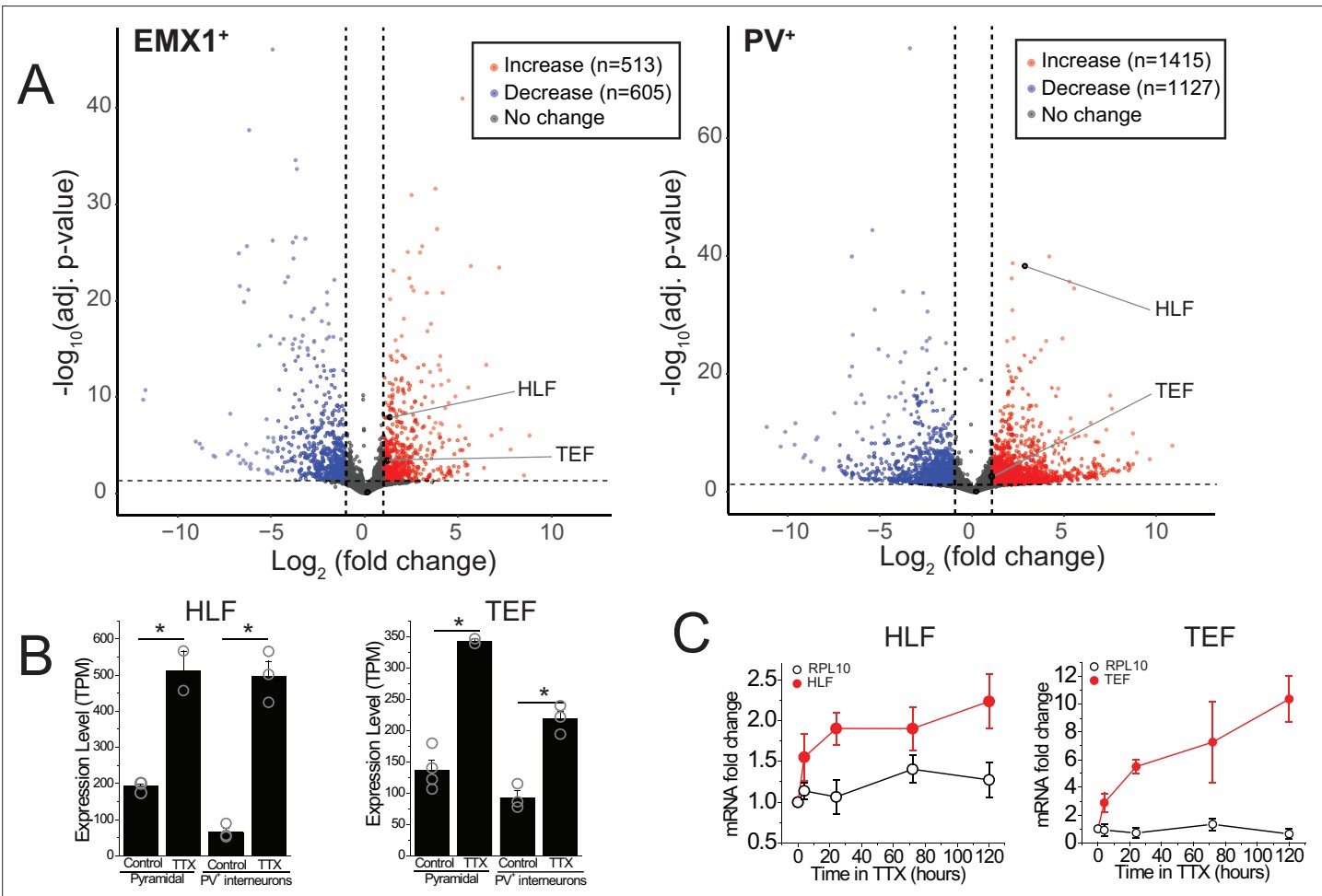

**Figure 1.** RNA-sequencing identifies transcripts affected by activity deprivation including the PARbZIP family of transcription factors. (**A**) Volcano plots of bulk RNA-seq from sorted fluorescently labeled pyramidal cells (left) and PV+ interneurons (right). Dashed lines indicate a fold change of 2 and adjusted p value of 0.05. Differential expression analysis revealed that TTX resulted in upregulation (513 in pyramidal, 1415 in PV+ interneurons) as well as downregulation (605 in pyramidal, 1127 in PV+ interneurons) of genes, among which *Hlf* and *Tef* were identified as upregulated with activity block in both excitatory and inhibitory neurons. Statistical analysis (Wald test followed by Benjamini-Hochberg correction) reveals that *Hlf* and *Tef* are significantly upregulated in both pyramidal cells and PV+ interneurons, while *Dbp* and *Nfil3* are not significantly altered in either cell type. (**B**) Bar graphs displaying transcript per million (TPM) values of *Hlf* (left panel) and *Tef* (right panel) in pyramidal cells (left) and interneurons (right) in control and 5-day TTX treated slices. Bars are mean values +/- SEM, open symbols are individual experiments. (**C**). Quantitative real-time PCR analysis of *Hlf* and *Tef* expression in whole cortex lysates following a time course of activity deprivation (n = 3–4 slices per time point; error bars are SEM). Two-way ANOVA reveals significant differences between *Hlf/Tef* and RPL10 (p<0.05).

The online version of this article includes the following source data for figure 1:

**Source data 1.** qPCR Data.

5 days during an early developmental period (equivalent postnatal days, EP, 12–17). This manipulation induces a robust homeostatic program and profoundly changes network dynamics (*Koch et al., 2010*; *Schaukowitch et al., 2017*). To characterize the changes in gene expression, we sorted deep layer (L5 and L6) pyramidal and PV$^+$ interneurons and performed RNA sequencing to look for transcripts that change following prolonged silencing. Inactivity activates a robust transcriptional program of both positively and negatively affected genes (*Figure 1A*) consistent with data from dissociated hippocampal cultures (*Schaukowitch et al., 2017*). To identify transcriptional regulators of homeostatic plasticity, we looked for TFs upregulated in the TTX condition. TFs can be classified into families, and in some cases subfamilies, which share a DNA binding domain, and so are expected to bind the same target DNA sequences in the genome and therefore to regulate the same or similar target transcripts. We asked which families or subfamilies were most overrepresented among the TFs differentially expressed during activity blockade. The most overrepresented subfamily or family was the PARbZIP subfamily of TFs, a subtype of the CEBP-related family in the class of Basic leucine zipper factors (classification data from tfclass.bioinf.med.uni-goettingen.de; *Wingender et al., 2015*). This subfamily includes three transcriptional activators *Hlf*, *Tef*, and *Dbp,* and a transcriptional repressor, *Nfil3*. Expression levels of *Dbp* are much lower than those of *Hlf* and *Tef* (<3 TPM in excitatory and PV$^+$ inhibitory neurons), suggesting that this family member may play a more limited role in the neocortex. Both *Hlf* and *Tef* are robustly upregulated after 5 days of activity deprivation in both excitatory (fold change [FC] TTX/control; FC = 2.6, 2.5; adjusted p value; $p_{adj}$ = 1.3e−8, 3.8e−4,) and inhibitory neurons (FC = 7.0, 2.1; $p_{adj}$ = 4.6e−39, 2.0e−3), while *Nfil3* and *Dbp* were not (FC = 0.51, 1.9; $p_{adj}$ = 0.49, 0.80, excitatory; FC = 1.2, 1.4; $p_{adj}$ = 0.84, 0.79 inhibitory).

To validate this finding and to further dissect the time course of PARbZIP TFs expression, we isolated the whole cortex from slice cultures and measured *Hlf* and *Tef* transcript levels at various time points during activity deprivation using real-time quantitative PCR. We find that upregulation occurs soon after activity withdrawal, since by 4 hr of TTX application the level of both TFs is already substantially elevated (*Figure 1C*), consistent with previous findings in dissociated hippocampal cultures (*Schaukowitch et al., 2017*). We also find that *Hlf* and *Tef* continue to increase with time, suggesting that their upregulation correlates with how long the network has been silenced.

## PARbZIP TFs restrain network homeostatic plasticity

Since *Hlf* and *Tef* are progressively upregulated during prolonged TTX treatment, which can lead to subsequent epilepsy (*Galvan et al., 2000*; *Scharfman, 2002*), we entertained the hypothesis that they may contribute to the homeostatic increase in network excitability following activity deprivation. On the other hand, loss of function of members of the PARbZIP family of TFs are also associated with epileptic phenotypes (*Hawkins and Kearney, 2016*; *Hawkins and Kearney, 2012*; *Rambousek et al., 2020*). Specifically, triple knockout (TKO) *Hlf$^{-/-}$/Dbp$^{-/-}$/Tef$^{-/-}$* mice have epilepsy (*Gachon et al., 2004*). One possibility that reconciles this apparent conflict is that instead of driving homeostatic plasticity, *Hlf* and *Tef* are upregulated in the TTX condition to restrain homeostatic responses. To test this hypothesis, we investigated the role that these TFs play in homeostatic plasticity. We induced homeostatic upregulation of network function by 2-day TTX application in organotypic slice cultures and measured network activity after TTX removal.

As a read-out of network activity, we measured changes in intracellular calcium using virally delivered GCaMP6f in primary somatosensory cortex (*Figure 2A*). In untreated slices at baseline, the neurons display population activity organized into complex, infrequent upstates which are similar to those observed in vivo (*Figure 2B*, *Sanchez-Vives and McCormick, 2000*). To quantify network activity, we measured the frequency of Ca$^{2+}$ peaks in each recording and the synchrony of calcium transients across cells (see Methods for detailed description of quantitative analysis). After 2 days of TTX treatment, the frequency of the upstates increases. While 2 days of TTX incubation increases the peak frequency 3.6-fold in wild-type (WT) slices (from 0.06 ± 0.01 to 0.22 ± 0.04 Hz), the same silencing duration produces a nearly sixfold increase in the TKO slices (from 0.09 ± 0.02 to 0.5 ± 0.05 Hz; *Figure 2*, *Figure 2—figure supplement 1A*). A two-way ANOVA revealed significant effects of treatment (TTX vs. Control; p<0.0001) and genotype (p<0.0001) and a significant interaction between the two (p<0.0001). Post hoc t tests (with Tukey correction for multiple comparisons) revealed that TTX increases in activity were highly significant in both WT (p=0.0007) and TKO (p<0.0001) slices and that TTX produced a significantly stronger increase in the TKO (p<0.0001),

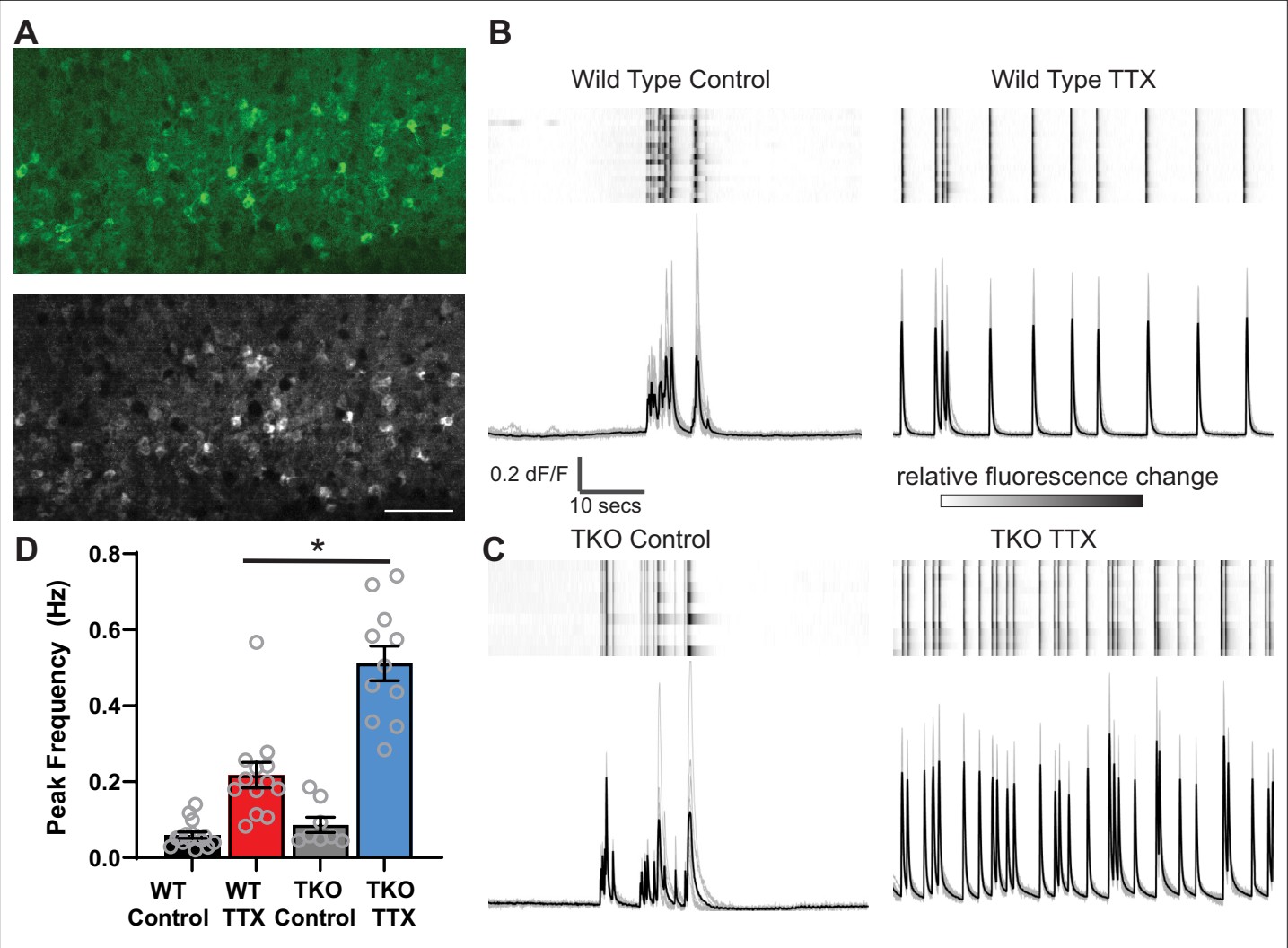

**Figure 2.** Calcium imaging reveals exaggerated response to activity deprivation in $Hlf^{-/-}/Dbp^{-/-}/Tef^{-/-}$ triple knockout (TKO) slices. (**A**). A confocal image of GCaMP6f fluorescence during an upstate in wild-type (WT) control slice (top panel) and a standard deviation projection of the same field of view of calcium signal during 1 min of recording (bottom panel). Scale bar = 50 μm. ROIs were manually selected around active cells identified by high standard deviation values. (**B, C**) GCaMP6f fluorescence heat map of selected ROIs (each row is one cell) during 1 min of recording (top panels) also shown as overlapping traces (gray, bottom panels) and an average fluorescence trace (black, bottom panels) in WT (**B**) and TKO slices (**C**) from control (left) and TTX-treated conditions (right). Traces show absolute dF/F, heatmap scale is normalized for each cell so that the full intensity range corresponds to the range of dF/F. (**D**) Quantification of peak frequency in GCaMP6f fluorescence traces. N = 16 slices for WT Control, N = 13 WT TTX, N = 8 TKO Control, and N = 11 TKO TTX. Error bars indicate SEM. TTX treatment significantly increases peak frequency in WT slices with significant (p<0.0001) interaction between treatment and genotype. The increase is more dramatic in the TKO slices. Two-way ANOVA with Tukey's correction for multiple comparisons to test statistical significance between conditions *p≤0.0001.

The online version of this article includes the following source data and figure supplement(s) for figure 2:

**Source data 1.** Calcium imaging analysis summary, 2D TTX; WT, TKO.

**Figure supplement 1.** Ca²⁺ activity in mutant slices is similarly exaggerated after 2 and 5 days of activity withdrawal.

**Figure supplement 1—source data 1.** Calcium imaging analysis summary, 5D TTX; WT, TKO.

**Figure supplement 2.** Ca²⁺ activity recovers to baseline levels following washout of TTX in both wild-type and mutant slices.

**Figure supplement 2—source data 1.** Calcium imaging analysis summary, recovery.

while the baseline activity of cells from slices derived from the TKO and WT do not differ significantly (p=0.92). Consistent with this, normalizing the TTX response to the control response revealed a larger relative increase in TKO than WT cultures (***Figure 2—figure supplement 1A***). The TTX condition also increased synchrony (from 0.76 ± 0.03 to 0.95 ± 0.01) in WT, and in the TKO (from 0.74 ±

0.03 to 0.94 ± 0.05; *Figure 2—figure supplement 1B*). While the TTX effects were highly significant (two-way ANOVA; p<0.0001), there was no significant effect of genotype (p=0.43) and no interaction between treatment and genotype (p=0.65), presumably reflecting the fact that synchrony is already saturated by TTX treatment in the WT and cannot further increase in the TKO, but also indicating that baseline synchrony is not altered in the TKO. A power test revealed that with the effect size and variances observed, 95 slice cultures in each group would be needed to detect a significant difference between WT and TKO peak frequency, while for the synchrony measure, the required N would be 1609. These data suggest that the PARbZIP family of TFs normally functions to restrain homeostatic plasticity, and in their absence, the homeostatic response is exaggerated. Moreover, even though the mutation is associated with epilepsy, it does not make cortical neuronal circuits more excitable at baseline.

## Recovery from activity deprivation is unperturbed in the mutant slices

Homeostatic plasticity is thought to provide flexibility to the network and thus most forms are reversed after reintroduction of activity (*Desai et al., 2002*; *Hobbiss et al., 2018*; *Koch et al., 2010*; *Wallace and Bear, 2004*). Reversal involves a homeostatic reduction of abnormally elevated activity. Even when this involves symmetric opposing changes in the same biophysical parameters altered in the response to activity deprivation (such as with downscaling and upscaling of excitatory quantal amplitudes), the mechanisms involved typically differ (*Stellwagen and Malenka, 2006*; *Sun and Turrigiano, 2011*; *Tan et al., 2015*; *Wang et al., 2017*). In addition, changes in activity following rebound from deprivation could reflect either a change in the vigor of the circuit's attempt to restore activity (i.e., enhanced homeostatic plasticity) or a persistent change in the setpoint to which activity is returned (*Styr et al., 2019*), or both. To test whether the transcriptional restraint of homeostasis is required for bidirectional flexibility and to assess whether the setpoint had changed, we asked whether the network is able to return back to baseline activity levels when the activity blockade is removed. We measured network activity immediately after the end of silencing with TTX as well as following two days of recovery in TTX-free media. In WT slices, activity deprivation causes hyperactivity, evident from a twofold rise in the peak frequency of the calcium activity. However, this exuberant activity is restored back to baseline levels within 2 days following restoration of action potential firing. In the TKO slices, even though the initial response to TTX is exaggerated, the network is also able to return to baseline levels following 2 days of recovery, to levels indistinguishable from WT (*Figure 2—figure supplement 2*). These data suggest that the recovery from a high activity state is not diminished in the TKO and suggest that the mutation in fact produces exaggerated homeostatic plasticity rather than a change in activity setpoint.

## Frequency, but not amplitude, of mEPSCs is disproportionately upregulated by deprivation in TKO slices

Network activity is unaltered in the mutant at baseline but the response to activity deprivation is exaggerated. To learn more about the mechanism by which *Hlf* and *Tef* restrain homeostatic plasticity, we probed the effects of TTX on excitatory synaptic transmission. Because of the high levels of spontaneous network activity, it was not feasible to study unitary action potential evoked transmission. Instead, to broadly assay the properties of excitatory synapses, we measured pharmacologically isolated AMPA-receptor driven miniature EPSCs in control and TTX-treated slices. We observed a robust upregulation of mEPSC amplitude following 2 days of activity deprivation, consistent with our understanding of homeostatic synaptic scaling (*Turrigiano et al., 1998*). We also observed a large increase in mEPSC frequency, which has also been described in this preparation (*Koch et al., 2010*) and in multiple other preparations (*Wierenga et al., 2006*). In the TKO mice, the synaptic scaling is unaffected (*Figure 3A and C*; amplitude increased from 13.4 ± 1.1 to 18.5 ± 1.2 pA in WT and from 13.2 ± 1.4 to 19.7 ± 1.1 pA in TKO) but the increase in mEPSC frequency in response to activity deprivation is more dramatic (*Figure 3B and D*; 4.0 ± 0.8 to 6.2 ± 0.7 Hz in WT vs. 3.5 ± 0.5–9.3 ± 1.0 Hz in the KO). Notably, both the frequency and amplitude of mEPSCs are the same between WT and mutant slices at baseline which parallels our observations of network activity (*Figure 2*). These results suggest that the members of the PARbZIP family of TFs function to restrain upregulation of mEPSC frequency in response to activity deprivation and this increase is correlated with changes in activity levels across the network.

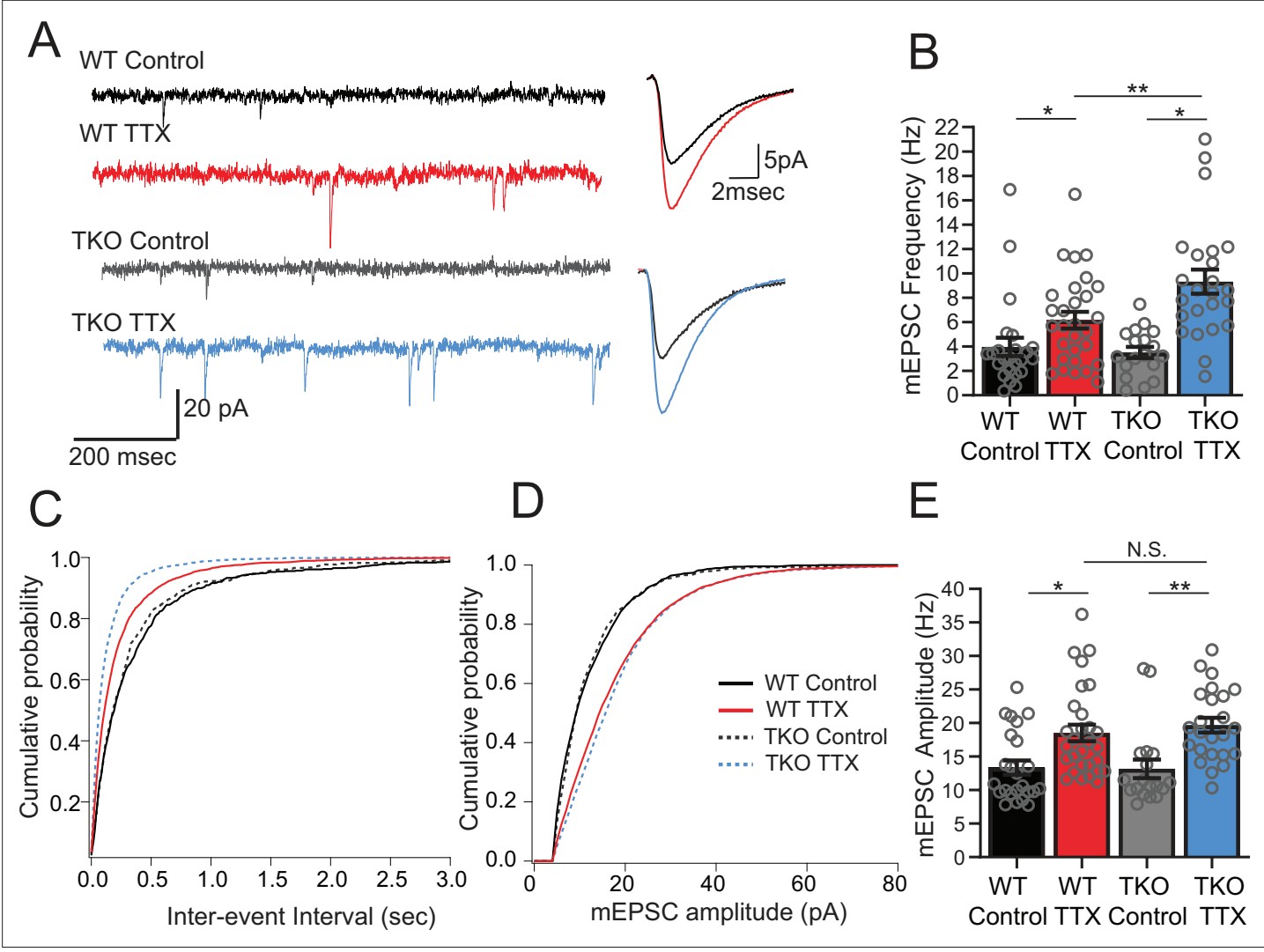

**Figure 3.** Frequency but not amplitude of excitatory synaptic currents is disproportionally upregulated in TTX-treated TKO slices. (**A**) Representative traces of mEPSC recordings from WT control (black), WT TTX-treated (red), TKO control (gray), and TKO TTX-treated (blue) slices. Right panel, average mEPSC waveforms for the same conditions. Both the frequency and the amplitude is increased in TTX-treated slices; however, the increase in frequency is more dramatic in TKO slices. (**B**) Quantification of mEPSC frequency for each condition. Colored bars with error bars are mean ± SEM, open circles are individual cells; N = 24 for WT control, N = 29 for WT TTX, N = 18 for TKO control, and N = 24 for TKO TTX. Two-way ANOVA revealed significant main effect of TTX treatment (p < 0.0001), genotype alone (p=0.030) and interaction between treatment and genotype (p = 0.041). Post hoc Tukey test revealed enhanced effect of TTX on mEPSC frequency in TKO cells compared to WT (p = 0.009, **). (**C**) Cumulative probability histogram for mEPSC inter-event intervals in each condition. (**D**) Cumulative probability histogram for mEPSC amplitudes in each condition. (**E**) mEPSC amplitudes for each condition. Two-way ANOVA revealed significant main effect for drug treatment (p < 0.0001) but not genotype (p = 0.74) or interaction between treatment and genotype (p = 0.55). TTX treatment enhanced mEPSC amplitude in both WT (p = 0.012, Tukey post hoc test, *) and TKO (p = 0.0035, Tukey post hoc test, **) to the same extent (p = 0.89, WT TTX compared to TKO TTX, Tukey post hoc test, N.S.). TKO, triple knockout; WT, wild-type.

The online version of this article includes the following source data for figure 3:

**Source data 1.** mEPSC amplitude and frequency.

## Neither the frequency nor amplitude of mIPSCs in pyramidal neurons in the TTX condition is affected in the TKO

Since L5 pyramidal neurons also receive inhibitory input, which is also subject to homeostatic plasticity (*Kilman et al., 2002*; *Kim and Alger, 2010*), we wanted to determine whether the response of mIPSCs to activity deprivation is also exaggerated in the mutant mice. In WT cultures, measurement of inhibitory input onto L5 pyramidal neurons revealed that both the amplitude and frequency of mIPSCs drop in response to activity deprivation (*Figure 4*). When we measured mIPSCs in pyramidal neurons

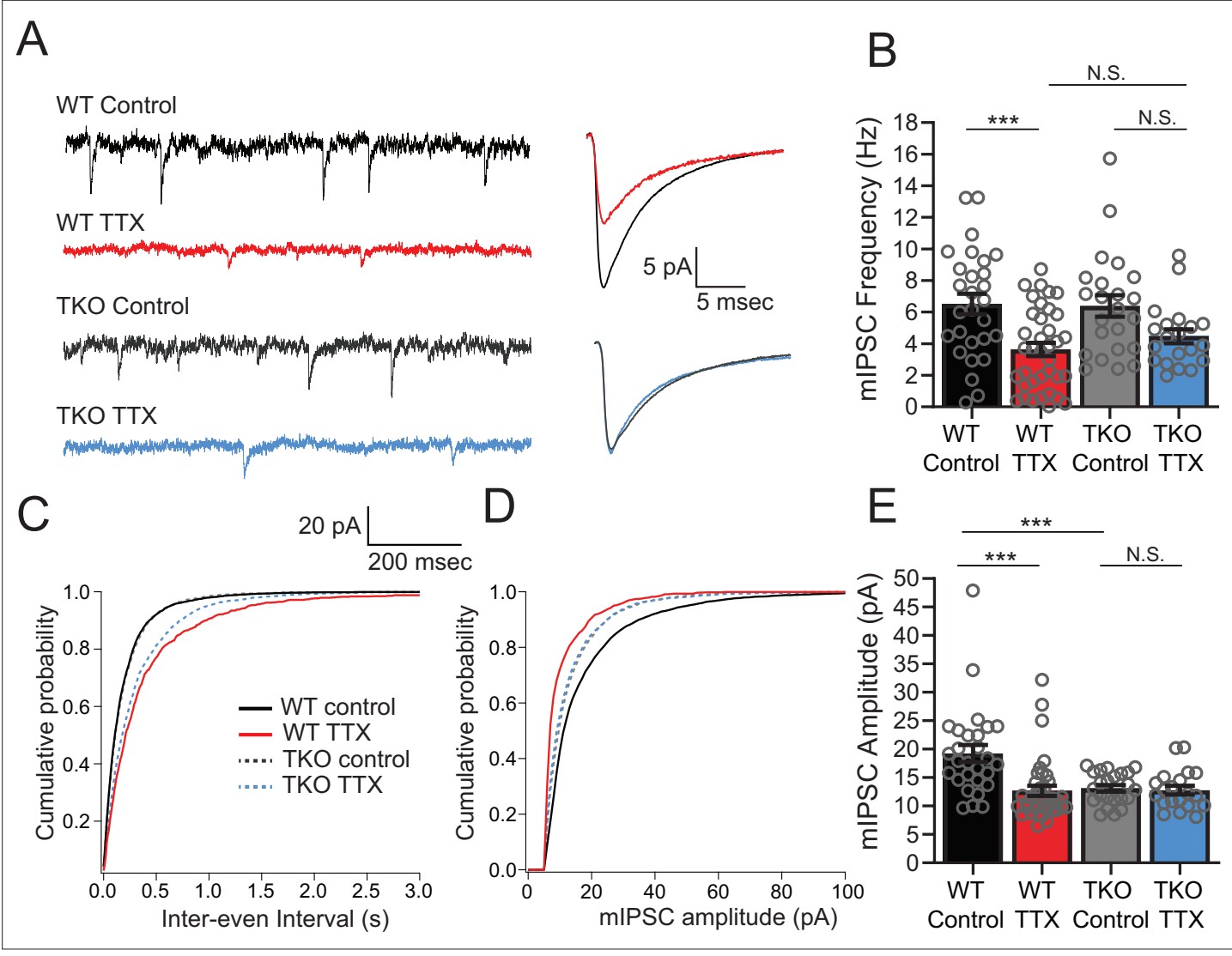

**Figure 4.** Inhibitory synaptic currents are not affected by the *Hlf$^{-/-}$/Dbp$^{-/-}$/Tef$^{-/-}$* mutation in the TTX condition. (**A**) Representative traces of mIPSC recordings from WT control (black), WT TTX-treated (red), TKO control (gray), and TKO TTX-treated (blue) slices. Right panel, average mIPSC waveforms for the same conditions. TTX treatment results in a diminished effect on mIPSC frequency in TKO and does not further decrease the amplitude of mIPSCs in TKO slices. (**B**) Quantification of mIPSC frequency. Colored bars with error bars are mean ± SEM, open circles are individual cells. Two-way ANOVA revealed a significant main effect of TTX treatment (p < 0.0001) but not genotype (p = 0.54). There was no significant interaction between treatment and genotype (p = 0.40), N = 28 for WT control, N = 37 for WT TTX, N = 23 for TKO control, and N = 20 for TKO TTX. TTX treatment decreased mIPSC frequency in WT (p = 0.0007, post hoc Tukey test) but not in TKO cells (p = 0.14, post hoc Tukey test). mIPSC frequency in TTX-treated cells is not different in TKO slices compared to WT (p = 0.72). The normalized change in frequency (TTX normalized to Control) was 0.55 in WT and 0.70 in TKO. We performed a power analysis that revealed that for the observed effect size and variance a sample of 68 neurons per condition would be required for significance. In (**B**), (**C**), and (**E**), cumulative probability histogram for mIPSC inter-event intervals in each condition. (**D**) Cumulative probability histogram for mIPSC amplitudes in each condition. (**E**) Quantification of mIPSC amplitudes for the same conditions as in (**B**). Two-way ANOVA revealed significant main effect of TTX (p = 0.002), and genotype (p = 0.0067), and a significant interaction between treatment and genotype (p = 0.005). TTX treatment decreased mIPSC amplitude in WT (p < 0.0001, Tukey post hoc test, ***), but not in TKO cells (p = 0.997, Tukey post hoc test, N.S.) presumably due to significantly lower baseline mIPSC amplitude in TKO slices compared to WT (p = 0008, post hoc Tukey test, ***). TKO, triple knockout; WT, wild-type.

The online version of this article includes the following source data for figure 4:

**Source data 1.** mIPSC amplitude and frequency.

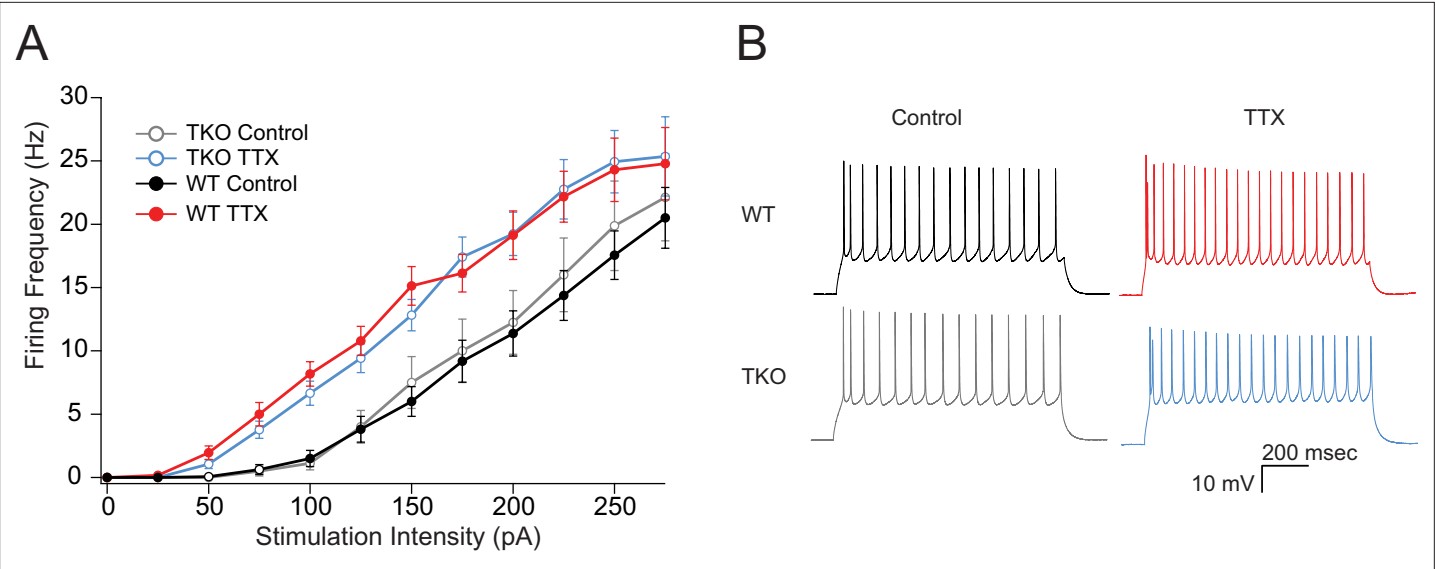

**Figure 5.** The effects of TTX on intrinsic excitability are not altered in TKO pyramidal neurons. (**A**) Comparisons of frequency-current relationships for layer five pyramidal neurons in control (black, closed circles) and TTX-treated (red, closed circles) WT slices, and in control (gray, open circles), and TTX-treated (blue, open circles) TKO slices, N = 27 cells for WT TTX, N = 16 for WT control, N = 8 for TKO control, and N = 17 for TKO TTX; error bars are SEM. Two-day TTX treatment increases intrinsic excitability in both WT and TKO slice cultures to a similar extent. A three-way mixed ANOVA with between subjects factors of treatment and genotype, and a within subjects factor of current level, revealed significant main effects for TTX treatment ($F_{(1,60)} = 18.1$, $p = 7.3e{-}05$) and current level ($F_{(11,660)} = 2.3e{-}190$) but not for genotype ($F_{(1,60)} = 0.06$, $p = 0.94$). There was a significant interaction between treatment and current level ($p = 0.06$) but not for interactions of genotype with current level ($p = 0.93$) or with both current level and treatment ($p = 0.99$). (**B**) Example traces of a train of action potentials from a pyramidal neuron in response to a 175 pA 0.5 s depolarizing current injection in control (left), TTX-treated (right), WT (top), or TKO (bottom) slices. TKO, triple knockout; WT, wild-type.

The online version of this article includes the following source data for figure 5:

**Source data 1.** Frequency-Current curves.

from TKO slices, we saw a similar decrease in frequency following 48 hr TTX treatment, although the amplitude of the change was smaller than in the WT (*Figure 4B and C*). In contrast, the mIPSC amplitude change is abolished: mIPSC amplitude is decreased at baseline compared to WT but did not show a further decrease after TTX treatment (*Figure 4D and E*). However, the decreased amplitude in the untreated condition does not correlate with the unchanged baseline network activity levels (*Figure 2*), suggesting that the decrease of mIPSCs amplitude at baseline may be insufficient on its own to cause network hyperexcitability. Thus, both the amplitude and frequency of mIPSCs received by pyramidal neurons after activity deprivation are indistinguishable from the WT in the TKO slices, suggesting that homeostatic plasticity of mIPSCs is not directly driving the changes in the network activity following TTX treatment. We cannot rule out the possibility that although baseline activity is normal, the baseline reduction in mIPSC amplitude contributes to subsequent network plasticity indirectly, by contributing to induction of the changes in EPSCs observed, or by inducing some form of compensation other than the physiological parameters measured.

## Intrinsic excitability of pyramidal neurons is unaffected in TKO slices

Changes in neuronal excitability have also been described as part of the homeostatic response to activity deprivation (*Desai et al., 1999*; *Karmarkar and Buonomano, 2006*; *Lambo and Turrigiano, 2013*; *Moore et al., 2018*; *Yoshimura and Rasband, 2014*). We wanted to test whether changes in intrinsic excitability could be contributing to the exaggerated homeostatic response in the TKO slices similarly to mEPSCs. WT L5 pyramidal neurons become more excitable following prolonged activity deprivation (*Figure 5*; *Desai et al., 1999*). To account for difference in the proportion of adapting and non-adapting neurons (*Hattox and Nelson, 2007*) causing apparent changes in intrinsic excitability due to uneven sampling of the two groups, we measured the adaptation ratio of each cell and analyzed an equal number of adapting and non-adapting cells in each genotype. Changes in excitability due to activity deprivation are accompanied by a shift in the average adaptation ratio of

the cells in both WT and TKO slices (control WT: 3.8 ± 0.16, TTX WT: 0.72 ± 0.22, TKO control: 0.46 ± 0.25, TKO TTX: 0.71 ± 0.33; two-way ANOVA significant difference between control and TTX and no significant difference between WT and TKO). Excitatory neurons in L5 in the slices prepared from TKO animals are equally excitable at baseline and also show upregulation of intrinsic excitability to the same extent (*Figure 5*). These results suggest that changes in intrinsic excitability are not driving the exaggerated network response to activity deprivation and this form of homeostatic plasticity is not under the control of the Par bZIP TF family.

## PARbZIP family members play unequal roles in restraining homeostatic plasticity

Expression of both *Hlf* and *Tef* increase in excitatory neurons following activity deprivation. Because all three Par bZIP family members form homo- and heterodimers and bind to the same PAR-response element (*Gachon, 2007*), they may be able to compensate for one another. To test whether *Hlf*, *Tef*, and *Dbp* work together to regulate homeostatic plasticity or if they have unequal contributions to restraining network response to activity deprivation, we measured the homeostatic response while varying gene copy number of each TF. In addition to the WT and TKO data described in *Figure 2* and *Figure 2—figure supplement 1*, we measured calcium transients in slice cultures treated with or without TTX from 97 additional animals with various combinations of 0, 1, or 2 alleles of *Hlf*, *Tef*, and *Dbp* (full details of each experiment given in associated data file). Rather than attempt to statistically test individual differences between each of 27 potential genotypes (Null, Het, wild-type for the three

**Table 1.** Multivariate linear models of the dependence of calcium peak frequency (PF) on genotype. Listed linear models were fit to peak frequencies of calcium transients measured from 87 control and 98 TTX-treated slices each from separate animals that differed in their genotype (null, heterozygote or wild-type) for each of the Hlf, Tef, and Dbp genes. The data (see attached file) includes the 88 experiments shown in *Figure 2* and *Figure 2—figure supplement 1* (WT and TKO) as well as 97 additional experiments (intermediate genotypes). Regressions were performed using the lm function in R where the variables Hlf, Tef, and Dbp represent the numbers of alleles of each gene (0, 1, or 2) and the variable (Hlf + Tef) represented the sum of the number of Hlf and Tef alleles. The column p > F reports the significance of each model assessed from an F test. For models that accounted for >10% of the variance in the data (adj $R^2$ > 0.1; bold values), the intercept ($b_0$) and coefficients ($b_1$…) are listed, along with the 790 probability that the coefficient is nonzero (t test).

| Model (TTX) | Adj $R^2$ | Pr > F | $b_0$ (Pr > t) | $b_1$ (Pr > t) | $b_2$ (Pr > t) | $b_3$ (Pr > t) |
|---|---|---|---|---|---|---|
| 1  PF = $b_0$ + $b_1$(*Hlf*) + $b_2$(*Tef*) + $b_3$(*Dbp*) | **0.284** | 2E−07 | 0.457 (<2E−16) | −0.055 (0.003) | −0.068 (4E−4) | 0.006 (0.71) |
| 2  PF = $b_0$ + $b_1$(*Hlf*) | **0.198** | 4E−06 | 0.423 (<2E−16) | −0.081 (4E−6) | | |
| 3  PF = $b_0$ + $b_1$(*Tef*) | **0.229** | 4E−07 | 0.437 (<2E−16) | −0.088 (4E−7) | | |
| 4  PF = $b_0$ + $b_1$(*Dbp*) | 0.073 | 0.007 | | | | |
| 5  PF = $b_0$ + b1(*Hlf*) + $b_2$(*Tef*) | **0.291** | 3E−08 | 0.458 (<2E−16) | −0.052 (0.003) | −0.066 (2E−4) | |
| 6  PF = $b_0$ + $b_1$(*Hlf* + *Tef*) | **0.296** | 4E−09 | 0.485 (<2E−16) | −0.059 (4E−9) | | |
| **Model (Ctrl)** | | | | | | |
| 7  PF = $b_0$ + $b_1$(*Hlf*) + $b_2$(*Tef*) + $b_3$(*Dbp*) | 0.044 | 0.08 | | | | |
| 8  PF = $b_0$ + $b_1$(*Hlf*) | −0.01 | 0.84 | | | | |
| 9  PF = $b_0$ + $b_1$(*Tef*) | 0.005 | 0.52 | | | | |
| 10  PF = $b_0$ + $b_1$(*Dbp*) | 0.063 | 0.05 | | | | |
| 11  PF = $b_0$ + $b_1$(*Hlf*) + $b_2$(*Tef*) | −0.01 | 0.62 | | | | |
| 12  PF = $b_0$ + $b_1$(*Hlf* + *Tef*) | −0.01 | 0.80 | | | | |

genes considered jointly), we fit a series of multivariate linear models to test the ability of genotype to predict calcium peak frequency. The same six models, described in *Table 1*, were fit separately to data from control and TTX-treated slices. The models, shown in *Table 1*, differ in how they depend on the number of alleles of each of the three genes. Model 1 depends on all three genes, models 2–4 depend only on one of the three genes (each in turn), and models 5 and 6 depend only on *Hlf* and *Tef*, either independently, or summed together. The ability of each model to account for the data was estimated from the adjusted $R^2$ which reflects the fraction of the variance in the data predicted by the model variables. Several key points were evident from this analysis:

1. All of the models produced poor fits (adjusted $R^2$ of –0.01 to 0.063) to the control data, consistent with the hypothesis that baseline activity does not depend on genotype.
2. All of the models that included variables for *Hlf* and/or *Tef* produced much better fits to the TTX data (adjusted $R^2$ of 0.198–0.296) consistent with the hypothesis that rebound activity is influenced by genotype. In these models, the coefficients (effect size) for *Hlf* and *Tef* were similar and t tests revealed they were highly significantly different from 0.
3. The model that only depended on *Dbp* (line 4) provided a poor fit to the TTX data (adjusted $R^2$ = 0.073) and in the model that included both terms for *Dbp* and the other two factors (line 1), the *Dbp* coefficient was small and not significantly different from 0. These observations are consistent with the hypothesis that rebound activity does not depend on *Dbp* and that even in a significant model including *Hlf* and *Tef*, *Dbp* does not add additional explanatory power to the model.
4. The models that included terms for both *Hlf* and *Tef* (line 5) or for their sum (line 6) produced the best fits (adjusted $R^2$ = 0.291 and 0.296); that is, better than those that depend on only one of these factors (lines 2 and 3; adjusted $R^2$ = 0.198 and 0.229). This is consistent with the hypothesis that both genes contribute to constraining cortical homeostatic plasticity.

Since the model with the highest adjusted $R^2$ was that which depended only on the sum of the number of *Hlf* and *Tef* alleles (line 6), we used this model for further post hoc (Tukey) tests to determine the effect of different numbers of alleles on the TTX response. The results of this analysis are shown in *Figure 6*. The most significant differences were between TKO slices (0 alleles; n = 32, mean = 0.48) and the other genotypes tested (1 allele, n = 16, mean = 0.32; 2 alleles, n = 26, mean = 0.35; and 4 alleles, n = 24, mean = 0.23; p < 0.0001–0.0014). Most of the other comparisons were not significant, with the exception of the difference between WT responses and those for animals with only two alleles (p = 0.011). This indicates that most of the difference between TKO (0 alleles of *Hlf* + *Tef*) and WT (4 alleles of *Hlf* + *Tef*) could be restored by a single allele, although there was a small but significant improvement between two and four alleles. The fact that this difference was significant, but the differences between 1 and 2 alleles, and 1 and 4 alleles were not, may reflect a real difference between these groups, or may reflect the imbalanced numbers of animals in each group. The group with two alleles included animals that were WT for *Hlf* but lacked *Tef* (N = 8), animals that were WT for *Tef* and lacked *Hlf* (N = 9), and trans-Het animals that were heterozygous for both *Hlf* and *Tef* (N = 7). A separate analysis of variance of these three subgroups revealed that the means (0.32, 0.41, and 0.33) were not significantly different (p = 0.28), consistent with the hypothesis that alleles of *Hlf* and *Tef* can substitute for one another in their ability to regulate the homeostatic response to activity deprivation.

To further dissect the relative contribution of the Par bZIP family members, we measured the survival of the mice with different numbers of copies of the TFs. TKO animals develop spontaneous seizures and have a dramatically decreased lifespan (*Gachon et al., 2004*). While our objective was not to make a detailed study of the seizure phenotype, we also observed, on multiple occasions, TKO animals that experienced spontaneous seizures, even though for most of cases, the instance of death was not directly observed. However, just one allele of either *Hlf* or *Tef* restores the lifespan of the animals to levels not statistically different from WT. Mice with one *Dbp* allele have improved survival compared to TKO animals but still die at a statistically significant higher rate than WT mice (*Figure 7*). These results differ from the cortical network activity response to activity deprivation in organotypic slices where the presence of *Dbp* had no effect on the constraint of homeostatic plasticity. This suggests that even though *Dbp* is dispensable for restraining homeostatic plasticity in the cortex, it may play a role in preventing premature death by functioning elsewhere in the brain or peripheral tissue (*Stewart et al., 2020*) since these TFs are expressed and play important roles there (*Wahlestedt et al., 2017*; *Wang et al., 2010*). However, a single allele of *Hlf* and *Tef* is enough to compensate for loss of the rest of the family members and restore survival to WT levels.

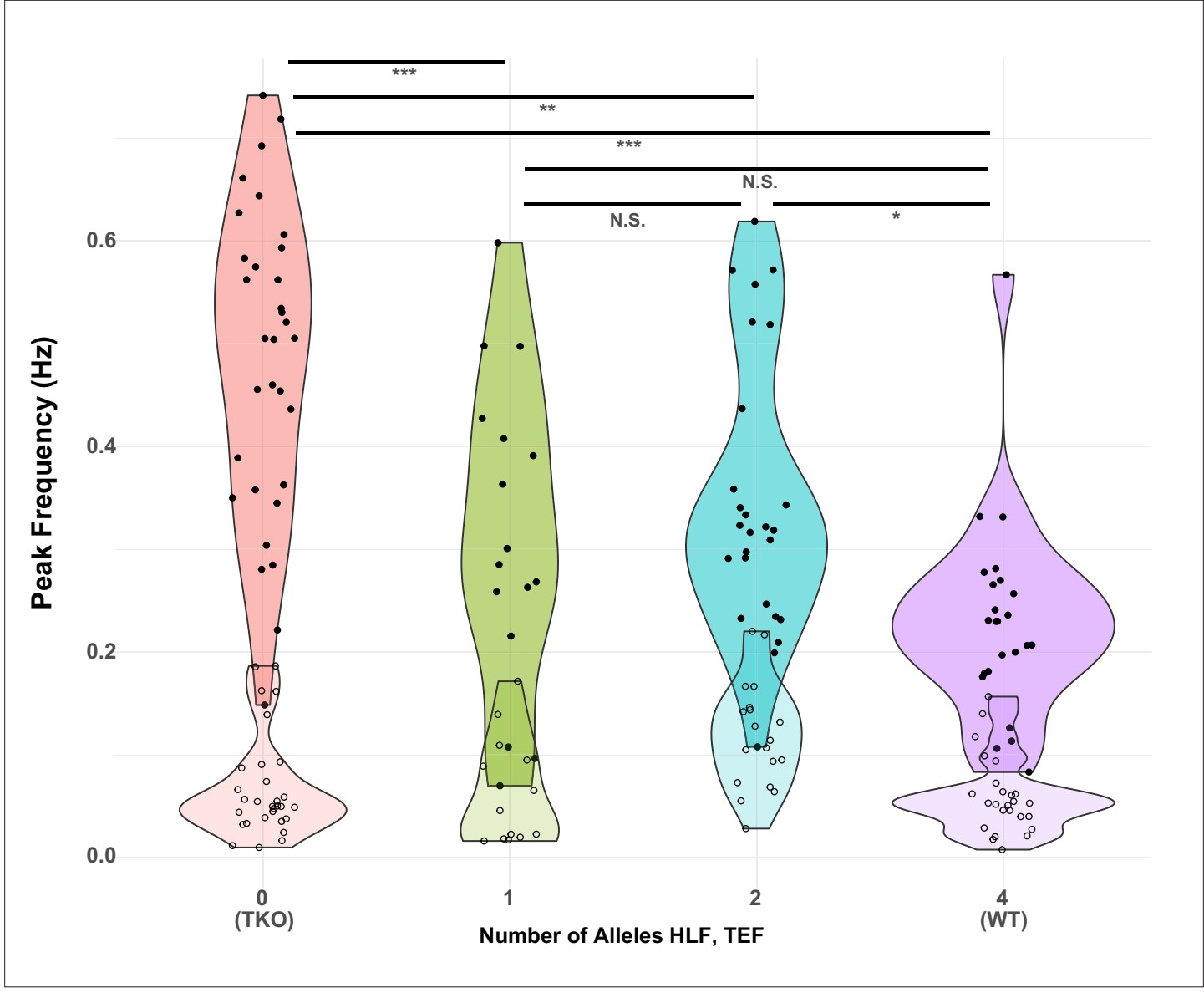

**Figure 6.** Presence of one allele of *Hlf* or *Tef* largely restores the WT response to activity deprivation. Violin plots show the distribution of peak frequencies measured from TTX-treated (upper, darker colors, filled symbols) and control (lower, lighter colors, open symbols) slice cultures made from animals carrying 0, 1, 2, or 4 alleles of the PARbZIP TFs *Hlf* and *Tef*. Data are those used in *Table 1* with the X-axis corresponding to models 6 (TTX) and 12 (Control). Horizontal positions of individual data points are jittered to prevent overplotting. Horizontal lines at top indicate results of post hoc t tests between levels for the TTX data with Tukey correction for multiple comparisons. *adj. $p < 0.05$, **adj. $p < 0.01$, ***adj. $p < 0.001$, N.S. – not significant, adj. $p > 0.05$. No post hoc testing was performed for the Control data since the fit of model 12 was not significant. WT, wild-type.

The online version of this article includes the following source data for figure 6:

**Source data 1.** Calcium imaging analysis summary, 2D, 5D TTX; all genotypes.

## Discussion

Inappropriately triggered homeostatic plasticity can either fail to compensate for changes in activity or can itself destabilize network activity (*Nelson and Valakh, 2015*). Although some molecular pathways that are required for the induction of homeostatic plasticity have been identified, whether and how homeostatic plasticity is negatively regulated is unknown. Here, we show that reduced activity, which initiates a compensatory increase in network excitability, also activates a set of TFs that function to restrain homeostasis. Thus, we propose a model in which homeostatic plasticity is negatively regulated through the PARbZIP family of TFs.

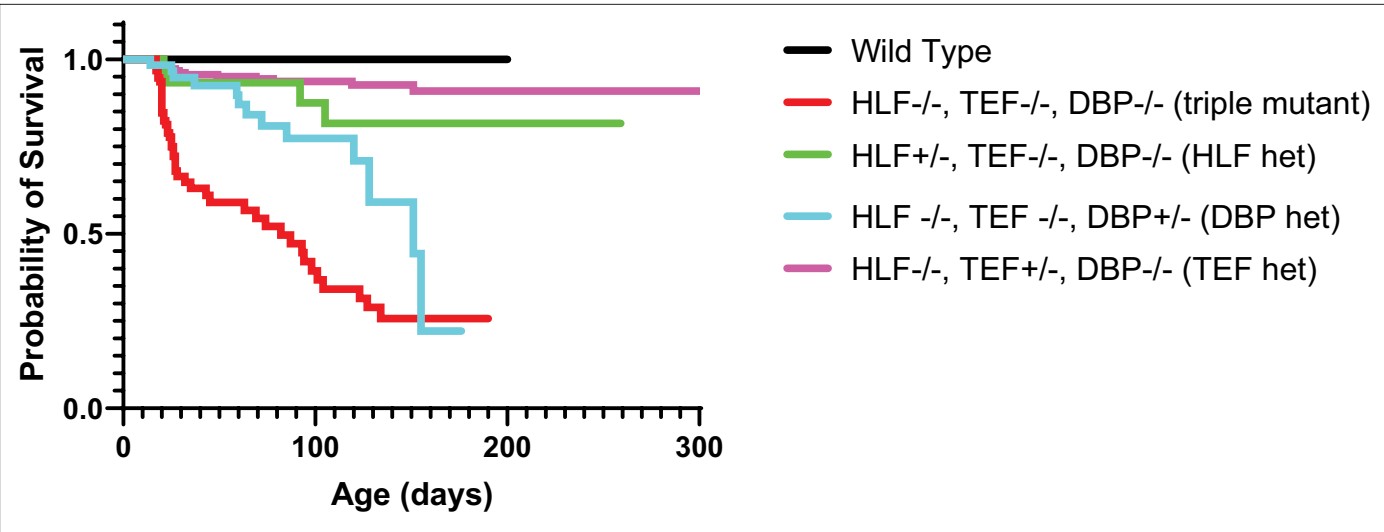

**Figure 7.** Presence of one allele of a PARbZIP TF improves survival relative to TKO animals. TKO mice have dramatically decreased survival (red line) compared to wild-type (WT) mice (black line). Presence of one copy of either *Hlf* (green line) or *Tef* (purple line) prevents premature death and the survival curves are not statistically different from WT, p = 0.12 and p = 0.56, respectively, Log-rank test with Bonferroni correction for multiple comparisons. DBP heterozygous mice (blue line) also have improved survival over TKO mice (p = 0.003) but are still statistically more likely to die prematurely than WT mice (p = 0.0003). N = 59 for DBP heterozygous, N = 96 for TKO, N = 36 for HLF heterozygous, N = 23 for WT, and N = 179 for TEF heterozygous. TKO, triple knockout.

The online version of this article includes the following source data for figure 7:

**Source data 1.** Survival data by genotype.

**Source data 2.** Statistical analysis of survival data.

Excitatory neurons respond to a global drop in network activity by upregulating the frequency and amplitude of excitatory synaptic currents, decreasing inhibitory currents, and increasing intrinsic excitability. These changes correlate with dramatically increased network activity immediately following removal of activity blockade that 'overshoots' initial activity levels. Synaptic scaling of excitatory synapses has previously been shown to depend on intact transcription and translation (*Dörrbaum et al., 2020*; *Goold and Nicoll, 2010*; *Ibata et al., 2008*; *Schanzenbächer et al., 2016*). Here, we demonstrate a second role for transcription in regulating the strength of the network homeostatic response. Loss of *Hlf*, *Tef*, and *Dbp* results in exaggerated homeostatic changes. However, the levels of network activity at baseline are unchanged. This suggests that these TFs are activated to restrain homeostatic signaling rather than downregulating network dynamics directly. While a few other genes have been described that have limited effects on network dynamics, but are required for homeostatic plasticity (Shank3, *Tatavarty et al., 2020*; FMRP, *Soden and Chen, 2010*; and Caspr2, *Fernandes et al., 2019*), this is, to our knowledge, the first negative regulator of homeostasis to be described.

Present data implicate mEPSC frequency, but not amplitude, as the site of the restraint of homeostatic plasticity. Although 2 days of activity blockade in dissociated cortical cultures were initially found to produce scaling up of EPSC amplitudes without accompanying changes in mEPSC frequency (*Turrigiano et al., 1998*), multiple subsequent studies have found evidence for changes in both amplitude and frequency as found here (*Echegoyen et al., 2007*; *Koch et al., 2010*; *Thiagarajan et al., 2005*; *Wierenga et al., 2006*). Changes in mini frequency are typically interpreted as reflecting increased presynaptic release, either via increases in the numbers of synapses or active release sites, or via changes in the rate at which spontaneous, action potential independent release occurs at each site. Although the increase observed in mEPSC frequency (1.7-fold greater in TKO than WT), was lower than the relative increase in activity levels, measured from the frequency of calcium transients (2.5-fold greater in TKO than WT at 5 days), these two measures need not be linearly related. First, mini frequency can be distinct from action potential evoked release (*Crawford et al., 2017*), and second, in a highly recurrent network, the relationship between firing and synaptic strength can be nonlinear (*Toyoizumi and Abbott, 2011*). We cannot eliminate the contribution of inhibitory neurons as a contributing factor to homeostatic plasticity restraint, since *Hlf* and *Tef* are

expressed in this neuronal type and are also upregulated by inactivity. Interneuron function may also be regulated by *Hlf* and *Tef* and this role may predispose the network to seizures. Dissecting the contribution of different cell types during different developmental windows will require generation of a conditional knockout and selective cell-type-specific disruption of individual TFs at different times in development.

mIPSCs received by excitatory neurons cannot explain the exaggerated network hyperexcitability caused by TTX incubation in TKO slices since neither the amplitude nor frequency of mIPSCs is different in the TTX condition. We note, however, that the amplitude of mIPSCs is decreased at baseline in the untreated cultures. Such decrease of the strength of inhibitory currents is insufficient on its own to induce network excitability, since baseline network activity was unchanged in the TKO, but it may make the network more vulnerable to future insults.

It is tempting to imagine a direct relationship between the exaggerated homeostatic plasticity in TKO cortical slices and the increased seizures and spontaneous death of TKO animals. The lack of constraint of homeostatic response can potentially explain, at least in part, the seizure phenotype in animals lacking all three TFs. However, partial genetic compensation of DBP for the survival phenotype but not the homeostatic plasticity phenotype as assayed by calcium imaging, suggests that the propensity to develop seizures and diminished survival have a more complex origin involving other brain regions and perhaps other organs of the body.

*Gachon et al., 2004* first reported that deletion of the PARbZIP proteins TEF, HLF, and DBP caused spontaneous and audiogenic seizures. They suggested that this may reflect effects on neurotransmitter metabolism caused by a reduction in pyridoxal phosphate (PLP) since pyridoxal kinase, which catalyzes the last step in the conversion of vitamin B6 into PLP is a target of the PARbZIP proteins. PLP is a required cofactor for decarboxylases and other enzymes involved in the synthesis and metabolism of GABA, glutamate, and the biogenic amine neurotransmitters including dopamine, serotonin, and histamine. Using HPLC, they found no changes in GABA or glutamate, but reduced levels of dopamine, serotonin, and histamine in the knockout animals. Although changes in amine transmitters could potentially contribute to phenotypes in vivo, they are unlikely to contribute to plasticity in cortical slice cultures which do not include the brainstem nuclei from which these projections arise. Reduced glutamate decarboxylase action could potentially have contributed to the observed baseline reduction in mIPSCs since heterozygous (*Lazarus et al., 2015*) and homozygous (*Lau and Murthy, 2012*) mutants of Gad1, the gene encoding the GAD67 isoform of glutamic acid decarboxylase, have reduced mIPSC amplitudes. However, this does not account for the enhanced homeostatic overshoot following activity blockade, since in the TKO animals, activity blockade did not produce any further reduction in inhibition, but produced robust changes at excitatory synapses.

HLF and TEF form homo- or heterodimers and recognize similar DNA sequences (*Gachon, 2007*). Here, we demonstrate that these TFs can powerfully compensate for the loss of other family members. We varied genotype for all three PARbZip factors and used multiple regression models to show that the peak frequency of calcium transients depends roughly equally on *Hlf* and *Tef* but does not appear to depend on *Dbp*. It also appears that a single allele of either *Hlf* or *Tef* in the complete absence of the other factors is sufficient to preserve the normal restraint of homeostatic plasticity, producing a response to activity deprivation that is not significantly different from WT. However, given the large number of genotypes to be tested, the limited number of animals tested may have led us to miss subtle difference that varies with the number of additional alleles, since activity in the presence of two alleles was significantly different (albeit with a low effect size) from that in WT. We also found, in agreement with prior findings (*Gachon et al., 2004*), that *Tef* and *Hlf* can robustly compensate for loss of each other and completely restore life span. Such robust compensation and functional redundancy of these TFs highlight their importance and may explain why they have not been identified in GWAS studies as risk factors for epilepsy or other neuropsychiatric diseases.

The mechanisms linking activity deprivation to *Tef* and *Hlf* activation were not directly addressed in this work, but seem likely to depend on reduced calcium influx. Prior studies have suggested alternately that scaling up of mEPSC amplitudes depends upon calcium influx through either T-type (*Schaukowitch et al., 2017*) or L-type (*Li et al., 2020*) calcium channels. The former pathway has also been found to depend on the activity-dependent TFs ELK1 and SRF. Activation of the constraining TFs *Tef* and *Hlf* could be secondary or delayed responses dependent on changes in the transcription and translation of other activity-dependent TFs, or may represent a primary response to activity

deprivation, analogous to, but in the opposite direction of the well-studied immediate early genes such as *Fos*, *Arc*, and others.

Besides their function in early embryonic development (**Gavriouchkina et al., 2010**; **Wahlestedt et al., 2017**), the PARbZip TFs have been extensively studied for their roles in circadian rhythms in flies, zebrafish, and rodents (**Cyran et al., 2003**; **Vatine et al., 2009**; **Weger et al., 2021**). The present finding that *Hlf* and *Tef* also act to restrain homeostatic plasticity is consistent with a recent report demonstrating that PARbZIP TFs are differentially expressed in human epileptogenic tissue (**Rambousek et al., 2020**). In addition, CLOCK, an upstream regulator of these TFs, is necessary in cortical excitatory neurons for maintaining normal network activity and leads to epilepsy when conditionally deleted in these neurons (**Li et al., 2017**). Multiple components of the molecular clock are robustly expressed in the neocortex (**Bering et al., 2018**; **Gachon et al., 2004**; **Kobayashi et al., 2015**), consistent with the idea that they might have an additional function in the cortex that is distinct from their circadian role in SCN. However, the role of core clock genes in homeostatic plasticity has not yet been explored. In this work, we expand the role of the PARbZIP family of TFs to include the negative regulation of homeostatic plasticity.

Why should homeostatic plasticity be subject to such dual 'push/pull' regulation? Perhaps because the changes in drive and excitability which must be buffered vary so widely during development. During the first few weeks of postnatal development in rodents, cortical neurons go from receiving few synapses to receiving thousands. This may require developmental downregulation of the strength of homeostatic plasticity. Neuropsychiatric diseases, such as monogenic causes of ASD, can trigger homeostatic changes in circuit properties which restore overall firing rates, but nonetheless have maladaptive effects on cortical function and flexibility (**Antoine et al., 2019**; **Nelson and Valakh, 2015**). Conversely, loss of function of genes giving rise to developmental disorders can cause failures of homeostatic plasticity (**Blackman et al., 2012**; **Genç et al., 2020**). Our findings provide a potential target for enhancing homeostatic plasticity in contexts where it is insufficient, or for downregulating it when it is maladaptive, and therefore provide an avenue for identifying how the positive and negative regulators of homeostasis interact to stabilize network activity.

# Materials and methods

**Key resources table**

| Reagent type (species) or resource | Designation | Source or reference | Identifiers | Additional information |
|---|---|---|---|---|
| Gene (*Mus musculus*) | *Hlf* | MGI | MGI:MGI:96108 | |
| Gene (Mm) | *Tef* | MGI | MGI:MGI:98663 | |
| Gene (Mm) | *Dbp* | MGI | MGI:MGI:94866 | |
| Strain, strain background (Mm) | *Emx1*-ires-Cre | Jackson Labs | JAX:005628 | |
| Strain, strain background (Mm) | *Pvalb*-ires-Cre | Jackson Labs | JAX:017320 | |
| Strain, strain background (Mm) | triple knockout (TKO) $Hlf^{-/-}/Dbp^{-/-}/Tef^{-/-}$; TKO | Jackson Labs | EM:02489 | |
| Strain, strain background (Mm) | Ai9 | Jackson Labs | JAX:007909 | |
| Cell line (include species here) | | | | |
| Transfected construct (include species here) | | | | |

*Continued on next page*

*Continued*

| Reagent type (species) or resource | Designation | Source or reference | Identifiers | Additional information |
|---|---|---|---|---|
| Biological sample (include species here) | | | | |
| Antibody | (Include host species and clonality) | | | (Include dilution) |
| Recombinant DNA reagent | AAV-GCaMP6f | Addgene | pAAV.Syn.GCaMP6f.WPRE.SV40 (AAV9 capsid serotype) 100837 | |
| Sequence-based reagent | RPL10 Forward | This paper | PCR primers | 5'-CACGGCAGAAACGAGACTTT-3' |
| Sequence-based reagent | RPL10 Reverse | This paper | PCR primers | 5'-CACGGACGATCCTATTGTCA-3' |
| Sequence-based reagent | GAPDH (forward) | This paper | PCR primers | 5'-TCAATGAAGGGGTCGTTGAT-3' |
| Sequence-based reagent | GAPDH (reverse) | This paper | PCR primers | 5'-CGTCCCGTAGACAAAATGGT-3' |
| Sequence-based reagent | HLF (forward) | This paper | PCR primers | 5'-CGGTCATGGATCTCAGCAG-3' |
| Sequence-based reagent | HLF (reverse) | This paper | PCR primers | 5'-GTACCTGGATGGTGTCAGGG-3' |
| Sequence-based reagent | TEF (forward) | This paper | PCR primers | 5'-GAGCATTCTTTGCCTTGGTC-3' |
| Sequence-based reagent | TEF (reverse) | This paper | PCR primers | 5'-GGATGGTCTTGTCCCAGATG-3' |
| Peptide, recombinant protein | | | | |
| Commercial assay or kit | Ovation RNA-seq system | Nugen | | |
| Commercial assay or kit | Illumina library quantification kit | KAPA biosystems | | |
| Commercial assay or kit | Picopure RNA isolation kit | Life Technologies | | |
| Commercial assay or kit | iScript cDNA Synthesis Kit | Bio-Rad | #1708891 | |
| Commercial assay or kit | RNA Clean and Concentrator kit | Zymo Research | R1014 | |
| Commercial assay or kit | SYBR Green Supermix | Bio-Rad | 1725270 | |
| Chemical compound, drug | Tetrototoxin (TTX) | Abcam | ab120055 | |
| Chemical compound, drug | DNQX | Sigma-Aldrich | D0540 | |
| Chemical compound, drug | APV | R&D Systems | 0106 | |
| Chemical compound, drug | Picrotoxin (PTX) | R&D Systems | 1128 | |

*Continued on next page*

*Continued*

| Reagent type (species) or resource | Designation | Source or reference | Identifiers | Additional information |
|---|---|---|---|---|
| Software, algorithm | Calcium Imaging Analysis Code | https://github.com/VH-Lab/vhlab-TwoPhoton-matlab; copy archived at **Van Hooser, 2023** | | |
| Software, algorithm | MatLab | Mathworks https://www.mathworks.com/products/matlab.html | RRID:SCR_001622 | |
| Software, algorithm | IgorPro | WaveMetrics https://www.wavemetrics.com/ | RRID:SCR_000325 | |
| Software, algorithm | GraphPad PRISM | GraphPad Software https://www.graphpad.com/ | RRID:SCR_002798 | |
| Software, algorithm | STAR | https://github.com/alexdobin/STAR; **Dobin, 2023** | | |
| Software, algorithm | featureCounts (Rsubread package) | https://bioconductor.org/packages/release/bioc/html/Rsubread.html | | |
| Software, algorithm | DESeq2 | https://bioconductor.org/packages/release/bioc/html/DESeq2.html | | |

## Animals

All procedures were approved by the Institutional Animal Care and Use Committee at Brandeis University (Protocol #20002), and conformed to the National Institutes of Health Guide for the Care and Use of Laboratory Animals. The initial RNAseq screen (*Figure 1*) was performed using the Cre-dependent tdTomato reporter strain Ai9 (*Madisen et al., 2010*) crossed with either parvalbumin-ires-Cre (*Hippenmeyer et al., 2005*), or Emx1-ires-Cre animals (*Gorski et al., 2002*) obtained from Jackson Labs. All other experiments were performed using WT (C57BL/6J) and previously published TKO *Hlf*$^{-/-}$/*Dbp*$^{-/-}$/*Tef*$^{-/-}$ animals (*Gachon et al., 2004*) which were acquired from the European Mouse Mutant Archive (stock EM:02489) as frozen embryos and propagated via IVF using a provider-recommended protocol. TKO mice were housed on a 12/12 light/dark cycle in a dedicated, climate-controlled facility. Cages were enriched with huts, chew sticks, and tubes. Food and water were available ad libitum, and animals were housed in groups of 2–4 after weaning at p21. Mice of both sexes were used for experiments.

## Organotypic slice culture

Organotypic slices were dissected from P6 to P8 pups. Animals were anesthetized with a ketamine (20 mg/mL), xylazine (2.5 mg/mL), and acepromazine (0.5 mg/mL) mixture (40 µL, via intraperitoneal injection), the brain was extracted, embedded in 2% agarose, and coronal slices containing primary somatosensory cortex were cut on a compresstome (Precisionary Instruments, Greenville, NC) to 300 µm in ice-chilled ACSF (126 mM NaCl, 25 mM NaHCO3, 3 mM KCl, 1 mM NaH$_2$PO$_4$, 25 mM dextrose, 2 mM CaCl$_2$, and 2 mM MgCl$_2$, 315–319 mOsm) with a ceramic blade and placed directly onto six-well Millipore Millicell cell culture inserts (Millipore Sigma PICM0RG50, Burlington, MA) over 1 mL warmed neuronal media (1× MEM (Millipore-Sigma), 1× GLUTAMAX (Gibco Thermo Fisher Scientific), 1 mM CaCl$_2$, 2 mM MgSO$_4$, 12.9 mM dextrose, 0.08% ascorbic acid, 18 mM NaHCO$_3$, 35 mM 1 M HEPES (pH 7.5), 1 µg/mL insulin and 20% Horse Serum (heat inactivated, Millipore Sigma,

Burlington, MA), pH 7.45 and 305 mOsm). Slices were placed in media containing 1× PenStrep (Gibco Thermo Fisher Scientific) and 50 µg/mL gentamicin (Millipore Sigma) for 24 hr and subsequent media changes were antibiotic-free. The slices were then grown at 35°C and 5% $CO_2$. Media were changed every other day to 1 mL of fresh media. For TTX treatment, media containing 500 nM TTX was added during the media change at EP 12. For TTX washout experiments, two additional media changes with TTX-free media were performed 5 min apart.

## RNA-sequencing
### Cell sorting
Slice cultures were converted into a single-cell suspension as previously described (*Sugino et al., 2006*), with some modifications. Organotypic slice cultures (n = 3 from three separate animals per condition) were placed in ice-cold, oxygenated ACSF with 1% FBS, and 5% Trehalose (*Saxena et al., 2012*), that had been 0.4 µm filtered, containing blockers (APV, DNQX, and TTX) to prevent exci-totoxicity, and gently removed from the membrane. The cortex was micro-dissected under a Leica MZ 16F fluorescent microscope and the tissue was placed in an oxygenated room temperature bath for 45 min, supplemented with 1 mg/mL type XIV protease (Sigma-Aldrich). Afterward, the tissue was moved back to the ACSF solution (without protease) and triturated with fire-polished Pasteur pipettes of successfully smaller diameters (~600, 300, and 150 µm). Samples were then sorted with a BD FACSAria Flow Cytometer. All isolated material for mRNA sequencing was harvested using the picopure RNA isolation kit (Life Technologies) and subjected to an one column DNAase digestion. mRNA libraries were prepared as previously described (*O'Toole et al., 2017*): amplifying with the Ovation RNA-seq system (Nugen), sonicating with a Covaris S 220 Shearing Device, constructing the libraries with the Ovation Rapid DR multiplex System (Nugen), and quantifying library concentration using the Illumina library quantification kit (KAPA biosystems). Samples were sequenced on either the Illumina Nextseq or Hiseq machines to a depth of ~25 million reads. Illumina sequencing adapters were trimmed from reads using cutadapt followed by mapping to the mm10 genome with STAR using the ENCODE Long RNA-Seq pipeline's parameters. Reads mapping to exons of known genes were quantified using featureCounts in the *Rsubread* package (*Liao et al., 2019*) and differential expression analysis was conducted using DESeq2 (*Love et al., 2014*). One of the three Emx1-cre TTX replicates was severely contaminated (high levels of nonneuronal genes and low enrichment of neuronal genes) and was excluded. Adjusted p values for each gene are reported from a Wald Test evaluating the significance of the coefficient representing the treatment group (TTX or Control) followed by the Benjamini-Hochberg correction.

## Quantitative real-time PCR
RNA was extracted from neocortical regions of slice cultures using TRIzol reagent followed by RNA Clean and Concentrator Kit (Zymo Research R1014). cDNA was synthesized with 0.5 µg of RNA using the cDNA Synthesis Kit (Bio-Rad 170-8891) using random hexamers to generate cDNA from total RNA. Quantitative real-time PCR was performed using Corbett Research RG-6000 Real-Time PCR Thermocycler with SYBR Green Supermix (Bio-Rad). The following primer sequences were used: RPL10 (forward) 5'-CACGGCAGAAACGAGACTTT-3', RPL10 (reverse) 5'-CACGGACGATCCTATTGTCA-3', GAPDH (forward) 5'-TCAATGAAGGGGTCGTTGAT-3' GAPDH (reverse) 5'-CGTCCCGTAGACAAAAT GGT-3', HLF (forward) 5'-CGGTCATGGATCTCAGCAG-3', HLF (reverse) 5'-GTACCTGGATGGTGTCA GGG-3', TEF (forward) 5'-GAGCATTCTTTGCCTTGGTC-3', and TEF (reverse) 5'-GGATGGTCTTGTC CCAGATG-3'.

## Electrophysiology
Organotypic slice cultures were cut out of the cell culture inserts along with the membrane using a scalpel blade. Slices were transferred to a thermo-regulated recording chamber and continuously perfused with oxygenated ACSF. Neurons were visualized on an Olympus upright epifluorescence microscope with 10× air and 40× water immersion objectives. Visually guided whole-cell patch-clamp recordings were made using near-infrared differential interference contrast microscopy. Recording pipettes of 3–5 MΩ resistance contained internal solution with the following concentrations. For excitatory currents and intrinsic excitability: (in mM) 20 KCl, 100 K-gluconate, 10 HEPES, 4 Mg-ATP, 0.3 Na-GTP, 10 Na-phosphocreatine, and 0.1% biocytin. For inhibitory currents: (in mM) 120 KCl, 10

HEPES, 4 Mg-ATP, 0.3 Na-GTP, 10 Na-phosphocreatine, and 0.1% biocytin. Recordings were collected using an AxoPatch 200B amplifier (Axon Instruments, Foster City, CA), filtered at 10 kHz and were not corrected for liquid junction potentials. Data were collected on a Dell computer using custom software running on Igor Pro (WaveMetrics, Lake Oswego, OR).

## Intrinsic excitability

Cells were selected at random from layer 5 in primary somatosensory cortex. Whole-cell recordings were performed with a K-Gluconate-based internal recording solution. Synaptic currents were blocked by adding picrotoxin (PTX) at 25 µM, 6,7-dinitroquinoxaline-2,3-dione (DNQX) 25 µM, and (2R)-amino-5-phosphonovaleric acid (APV) at 35 µM to standard ACSF to block γ-aminobutyric acid (GABA), α-amino-3-hydroxy-5-methyl-4-isoxazolepropionic acid (AMPA), and N-methyl-d-aspartate (NMDA) receptors, respectively. Cells were held at –65 mV by injecting a small current. 500 ms current injections ranging from –25 to 275 pA in 25 pA steps increments at random intensity were delivered every 10 s. Adaptation ratio was calculated using custom-written IGOR script as previously described (**Hattox and Nelson, 2007**). Briefly, we calculated the ratio between the third and last inter-spike interval at two times the threshold current.

## Synaptic currents

To isolated miniature excitatory AMPA-mediated currents 25 µM PTX, 35 µM APV, and 500 nM TTX were added to the extracellular recording solution. To isolate inhibitory currents, 20 µM DNQX, 35 µM APV, and 500 nM TTX were added to the extracellular solution. The slices were incubated in TTX only during active recording, which lasted no more than 30 min for each slice. Cells were clamped at –65 mV with a series resistance of below 20 MΩ with no compensation for the liquid junction potential. Ten 10-s traces were acquired for each cell. Threshold-based detection of mPSCs and action potentials was implemented using custom-written IGOR scripts equipped with routines for baseline subtraction, custom filtering, and measurements of intervals, amplitudes, and kinetic properties.

## Calcium imaging

To visualize calcium dynamics, 24 hr after slice preparation, 1 µL of pAAV.Syn.GCaMP6f.WPRE.SV40 (AAV9 capsid serotype, Addgene 100837) was applied to the somatosensory cortex in each hemisphere. At day EP14, slices were cut out along with the membrane and placed into the perfusion chamber mounted on a spinning disc microscope (Leica DMI 6000B; Leica Microsystems, Inc, Buffalo Grove, IL), with an Andor CSU W1 spinning disc unit, using an Andor Neo sCMOS cameras and Andor IQ3 to run the system (Andor Technology PLC, Belfast, N. Ireland). Slices were allowed to acclimate while being perfused with 33°C oxygenated ACSF with no TTX for 10 min. Slices were imaged with a 10× objective centered over the primary somatosensory area. For each slice, 10 min of imaging from a 500-µm square per slice were acquired at 33 frames per second.

## Calcium imaging analysis

Cellular somatic ROIs were selected by hand from background using custom MATLAB code (http://github.com/VH-Lab/vhlab-TwoPhoton-matlab) (**Van Hooser, 2023**) based on a standard deviation projection of the video (**Figure 1A**, bottom panel). Cells were only considered viable if they: were of a characteristic size and shape for neurons; were bright enough to be clearly distinguished from background; were in focus with clear and sharp edges; and had several bouts of firing within the first minute. Active cells (n = 10–100) were selected by the experimenter within our area of imaging. We confirmed by subsampling cells in a subset of WT control experiments with more cells (n = 7 slices with 27–43 cells) that for the lowest sample of cells measured (n = 10), the estimates of peak frequency and synchrony had a coefficient of variation of the mean (SEM/mean calculated across sub-samples within a slice) of 0.05 and 0.01. From these ROIs, mean intensity per cell at each frame was computed. The raw intensity values were transformed to ΔF/F by diving by baseline values obtained either where the coefficient of variation within 50 frames was below 0.3 ΔF/F units/frame, or from the 5th percentile of all frames. Since the distribution of event amplitudes across cells was potentially dependent on other experimental factors such as the amount of calcium indicator expressed as well as optical factors such as the depth of the cell in the slice, we decided not to try to infer the magnitude of firing from the amplitude of events, but only periods of elevated firing from the timing of events. To produce a peak

frequency measure, noise was reduced by fitting the ΔF/F data with a smoothing spline, and active periods were defined as those with a slope above 0.30 ΔF/F units/frame that were within 30 frames of other frames with positive slope. The number of active periods reflected the number of times each cell was active, though this measure had no ability to predict the number of action potentials that made up each firing period. We excluded cells that failed to fire at any time during our period of recording. In addition, some of our slices exhibited some dimming of signal due to a combination of drift in the Z-dimension and fluorescent photobleaching. Experiments were discarded if this decrement caused firing peaks to fall and remain below the slope threshold in the selected cells. Synchrony was defined as the cross-correlation at delta T = 0 during upstates computed for all pairs of recorded cells and averaged across cells and across upstates.

## Acknowledgements

This research was supported by the National Institute of Neurological Disease and Stroke and by the Simons Foundation for Autism Research.

## Additional information

### Competing interests

Sacha B Nelson: Reviewing editor, *eLife*. The other authors declare that no competing interests exist.

### Funding

| Funder | Grant reference number | Author |
| --- | --- | --- |
| National Institute of Neurological Disorders and Stroke | R01NS109916 | Sacha B Nelson |
| Simons Foundation Autism Research Initiative | 648651 | Sacha B Nelson |

The funders had no role in study design, data collection and interpretation, or the decision to submit the work for publication.

### Author contributions

Vera Valakh, Conceptualization, Data curation, Formal analysis, Validation, Investigation, Visualization, Methodology, Writing - original draft, Project administration, Writing - review and editing; Derek Wise, Data curation, Software, Formal analysis, Investigation, Visualization, Methodology, Writing - review and editing; Xiaoyue Aelita Zhu, Data curation, Software, Formal analysis, Validation, Investigation, Visualization, Methodology, Writing - review and editing; Mingqi Sha, Jaidyn Fok, Data curation, Formal analysis, Investigation, Writing - review and editing; Stephen D Van Hooser, Software, Formal analysis, Writing - review and editing; Robin Schectman, Data curation, Formal analysis, Validation, Investigation, Writing - review and editing; Isabel Cepeda, Data curation, Formal analysis, Validation, Investigation, Visualization, Writing - review and editing; Ryan Kirk, Data curation, Software, Formal analysis, Validation, Visualization, Writing - review and editing; Sean M O'Toole, Formal analysis, Investigation, Writing - review and editing; Sacha B Nelson, Conceptualization, Resources, Data curation, Software, Formal analysis, Supervision, Funding acquisition, Visualization, Writing - original draft, Project administration, Writing - review and editing

### Author ORCIDs

Vera Valakh http://orcid.org/0000-0001-7149-1562
Stephen D Van Hooser http://orcid.org/0000-0002-1112-5832
Ryan Kirk http://orcid.org/0009-0002-6735-2513
Sacha B Nelson http://orcid.org/0000-0002-0108-8599

### Ethics

All procedures were approved by the Institutional Animal Care and Use Committee at Brandeis University (Protocol #20002), and conformed to the National Institutes of Health Guide for the Care and Use of Laboratory Animals.

### Decision letter and Author response

Decision letter https://doi.org/10.7554/eLife.74899.sa1
Author response https://doi.org/10.7554/eLife.74899.sa2

## Additional files

### Supplementary files

• MDAR checklist

### Data availability

All data generated or analyzed during this study are included in the manuscript and supporting files, calcium imaging analysis code is available https://github.com/VH-Lab/vhlab-TwoPhoton-matlab (copy archived at *Van Hooser, 2023*). RNAseq data have been deposited to the BioSample database under accession numbers: SAMN31104764, SAMN31104765, SAMN31104766, SAMN31104767, SAMN31104768, SAMN31104769, SAMN31104770, SAMN31104771, SAMN31104772, SAMN31104773, SAMN31104774, and SAMN31104775.

The following datasets were generated:

| Author(s) | Year | Dataset title | Dataset URL | Database and Identifier |
|---|---|---|---|---|
| Nelson S, Velakh V, O'Toole S, Kirk R | 2022 | A transcriptional constraint mechanism limits the homeostatic response to activity deprivation in mammalian neocortex | https://www.ncbi.nlm.nih.gov/biosample/?term=SAMN31104764 | NCBI BioSample, SAMN31104764 |
| Nelson S, Velakh V, O'Toole S, Kirk R | 2022 | A transcriptional constraint mechanism limits the homeostatic response to activity deprivation in mammalian neocortex | https://www.ncbi.nlm.nih.gov/biosample/?term=SAMN31104765 | NCBI BioSample, SAMN31104765 |
| Nelson S, Velakh V, O'Toole S, Kirk R | 2022 | A transcriptional constraint mechanism limits the homeostatic response to activity deprivation in mammalian neocortex | https://www.ncbi.nlm.nih.gov/biosample/?term=SAMN31104766 | NCBI BioSample, SAMN31104766 |
| Nelson S, Velakh V, O'Toole S, Kirk R | 2022 | A transcriptional constraint mechanism limits the homeostatic response to activity deprivation in mammalian neocortex | https://www.ncbi.nlm.nih.gov/biosample/?term=SAMN31104767 | NCBI BioSample, SAMN31104767 |
| Nelson S, Velakh V, O'Toole S, Kirk R | 2022 | A transcriptional constraint mechanism limits the homeostatic response to activity deprivation in mammalian neocortex | https://www.ncbi.nlm.nih.gov/biosample/?term=SAMN31104768 | NCBI BioSample, SAMN31104768 |
| Nelson S, Velakh V, O'Toole S, Kirk R | 2022 | A transcriptional constraint mechanism limits the homeostatic response to activity deprivation in mammalian neocortex | https://www.ncbi.nlm.nih.gov/biosample/?term=SAMN31104769 | NCBI BioSample, SAMN31104769 |

*Continued*

| Author(s) | Year | Dataset title | Dataset URL | Database and Identifier |
|---|---|---|---|---|
| Nelson S, Velakh V, O'Toole S, Kirk R | 2022 | A transcriptional constraint mechanism limits the homeostatic response to activity deprivation in mammalian neocortex | https://www.ncbi.nlm.nih.gov/biosample/?term=SAMN31104770 | NCBI BioSample, SAMN31104770 |
| Nelson S, Velakh V, O'Toole S, Kirk R | 2022 | A transcriptional constraint mechanism limits the homeostatic response to activity deprivation in mammalian neocortex | https://www.ncbi.nlm.nih.gov/biosample/?term=SAMN31104771 | NCBI BioSample, SAMN31104771 |
| Nelson S, Velakh V, O'Toole S, Kirk R | 2022 | A transcriptional constraint mechanism limits the homeostatic response to activity deprivation in mammalian neocortex | https://www.ncbi.nlm.nih.gov/biosample/?term=SAMN31104772 | NCBI BioSample, SAMN31104772 |
| Nelson S, Velakh V, O'Toole S, Kirk R | 2022 | A transcriptional constraint mechanism limits the homeostatic response to activity deprivation in mammalian neocortex | https://www.ncbi.nlm.nih.gov/biosample/?term=SAMN31104773 | NCBI BioSample, SAMN31104773 |
| Nelson S, Velakh V, O'Toole S, Kirk R | 2022 | A transcriptional constraint mechanism limits the homeostatic response to activity deprivation in mammalian neocortex | https://www.ncbi.nlm.nih.gov/biosample/?term=SAMN31104774 | NCBI BioSample, SAMN31104774 |
| Nelson S, Velakh S, O'Toole S, Kirk R | 2022 | A transcriptional constraint mechanism limits the homeostatic response to activity deprivation in mammalian neocortex | https://www.ncbi.nlm.nih.gov/biosample/?term=SAMN31104775 | NCBI BioSample, SAMN31104775 |

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
