## [Editor Report]

Homeostatic plasticity helps to maintain the stability of neural network activity. This study shows that activation of PAR bZIP family of transcription factors restrains homeostatic upregulation of network activity in response to activity deprivation in mouse brain slice cultures. The identification of an endogenous transcriptional program that limits upward homeostatic response and helps prevent aberrant activity associated with epilepsy and brain disorders is important, and the findings will be of interest to a broad neuroscience community.

---

## [Decision Letter]

**Decision letter after peer review:**

Thank you for submitting your article "A transcriptional constraint mechanism limits the homeostatic response to activity deprivation in mammalian neocortex" for consideration by *eLife*. Your article has been reviewed by 3 peer reviewers, including Yukiko Goda as Reviewing Editor and Reviewer #1, and the evaluation has been overseen by Gary Westbrook as the Senior Editor.

Essential revisions:

All three reviewers acknowledge that the study provides novel insights into a long-standing question about the mechanisms that constrain homeostatic synaptic plasticity, and the findings are exciting and of broad interest. However, a couple of major concerns have been raised regarding (a) the interpretation of calcium imaging data and (b) data quantification, effect sizes and statistics in a number of experiments.

(a) The main conclusion that PAR bZIP transcription factors constrain the network activity via regulating excitatory synapses should be further strengthened by addressing the following two points:

(i) Ruling out the contribution of changes in inhibitory synaptic transmission in TKO neurons. If the authors cannot rule out the contribution, then calcium imaging experiments should be conducted also in excitatory or inhibitory neurons to clarify the relative contributions.

(ii) In addition, calcium imaging, as the key physiology data, should be performed in WT and TKO cultures for two time points: after 2 days in TTX and after 5 days in TTX. Monitoring the kinetics will help in better relating the physiology data to the RNA-seq data and to clarify whether TKO cultures overshoots or if they show a faster compensation.

(b) The authors should fully address the concerns regarding data quantification and statistics that are raised by Reviewers 2 and 3.

*Reviewer #1 (Recommendations for the authors):*

1. The initial RNA seq analysis in Figure 1A,B compares control vs. TTX in L5-6 pyramidal and PV+ neurons. How well does the observation of strong upregulation of HFL and TEF extend to pyramidal and PV+ cells for a 2-day TTX treatment that is used for much of the later experiments? Notably, calcium imaging based on synapsin promoter-driven GCaMP6f expression does not specify cortical layers being targeted nor discriminate between excitatory and inhibitory neurons.

2. Figure 2. It would be informative to monitor calcium dynamics in a cell-type specific manner to decipher the differences in homeostatic response between WT and mutant neurons.

3. Line 266-269. It seems premature to conclude that homeostatic plasticity of mIPSCs is not driving the TTX-dependent change in network activity based on the fact that the levels of mIPSC amplitude and frequency are indistinguishable between WT and mutant after TTX treatment. In particular, the striking reduction in mIPSC amplitude in the baseline in the mutant prep indicates that already some form of compensation for altered mIPSCs has taken place to keep the network activity level comparable between WT and mutant prior to TTX treatment. In addition, the observed lack of a further change in mIPSCs could be due to saturation of mIPSC-dependent mechanisms.

*Reviewer #2 (Recommendations for the authors):*

I have concerns, mainly revolving around statistics and effect sizes:

1. Based on the RNA seq data shown in figure 1, it is unclear why the authors focused their analyses on Hlf, Tef and Dbp, because the data suggest more pronounced regulation of other genes. The authors should explain why these candidates were chosen for further analysis in the Results section.

2. One of the central findings of this study is that Hlf-/-/Dbp-/-/Tef-/- triple knockout (TKO) slice cultures display a more pronounced increase in calcium peak frequency upon TTX treatment compared to WT controls. In the Results section, it is mentioned that TTX treatment induces a two-fold increase in calcium peak frequency in WT, and a four-fold increase in TKO cultures (l. 142, 143). However, based on the mean data plotted in figure 2D, the increase in WT is around 3-4-fold (15 peaks x s^-1/4 peaks x s^-1=3.5), whereas the one in TKO is around four-fold (30/7.5=4). Could the authors comment on the discrepancy between the text and the data? Furthermore, it is key to test for an interaction between treatment and genotype to support this major conclusion by statistics. Additionally, it would be helpful to plot the relative increase in calcium peak frequency upon TTX treatment in WT and TKO.

3. On a similar note, although not statistically significant, the mean calcium peak frequency in TKO cultures is around two-fold higher than WT under baseline conditions (Figure 2D). A similar trend can be seen in the data shown in figure 6A (top panel), whereas the baseline calcium peak frequency is similar between TKO and controls in the experiment shown in Figure 2B. Based on the scatter of the data, can the authors exclude that baseline calcium peak frequency is increased in TKO cultures? I suggest running a power analysis for these data to verify if the sample sizes are sufficient to support their conclusions. Otherwise, sample sizes would have to be increased, or the conclusions should be revised/toned down. Alternatively, could additional parameters, such as inter-burst interval, be read out to support the conclusions that baseline network excitability is unchanged in TKO cultures, and that TTX application leads to a more pronounced increase in calcium peak frequency? Another possibility may be to pool the data of the different experiments?

4. Based on the data shown in figure 6, they conclude that one copy of Tef is sufficient to constrain homeostatic plasticity. Can they rule out that one copy of Hlf is sufficient as well? This would actually fit to the result that one copy of Tef or Hlf rescues lifespan in TKO. The peak frequencies of DKO;Tef +/- and DKO;Hlf +/- after TTX treatment actually look quite similar. Are these two groups statistically different from one another? Only comparing the absolute DKO;Tef +/- or DKO;Hlf +/- data to WT is not very informative. Again, I suggest normalizing the data of each genotype to the respective control, and to test for interactions between treatment and genotype.

5. Sample sizes (culture number/ if applicable, neuron number) should be clearly specified for each experiment. Are some data sets only based on two cultures (e.g. Figure 6 ctrl.)? Again, a power analysis may be helpful to determine sample sizes.

6. The major (recorded) effect underlying the increase in network excitability is an increase in mEPSC frequency. Does this indicate that the mechanisms that constrain homeostatic plasticity mainly work on presynaptic mechanisms, or an increase in synapse number? These possibilities should be also discussed.

7. TTX application increases mEPSC frequency by around two-fold and does not affect other recorded parameters, such as excitability. How could a two-fold increase in mEPSC frequency translate into a four-fold increase in calcium peak frequency? Is there any evidence for a supralinear increase in network activity with regard to miniature frequency? This issue should be discussed.

8. What is the evidence for 'exaggerated' homeostatic plasticity in TKO cultures? Can they rule out that TKO neurons compensate better? This possibility should be discussed unless they can exclude the possibility that TKO cultures indeed do not retarget set-point activity.

Other comments:

1. It would be very helpful if the abstract could contain more specific information (e.g. experimental system, what kind of homeostatic plasticity was studied, some specific results etc.)

2. The more pronounced increase in mEPSC frequency upon TTX treatment in TKO cultures compared to WT (Figure 3B) appears more robust than the corresponding changes in calcium peak frequency (Figure 2). However, they should also test for an interaction between treatment and genotype to support their conclusion by statistics for the data shown in figure 3B, C.

3. Similarly, although not statistically significant, the decrease in mIPSC frequency upon TTX treatment seems less pronounced in TKO cultures (Figure 4B, C). Could they run a power analysis and test for treatment-group interaction as well?

4. mIPSC amplitudes are decreased in TKO cultures, and there is no further decrease upon TTX application (Figure 4D, E). First, the decrease in baseline mIPSC amplitude in TKO is interesting, and may represent a homeostatic response that counteracts a potential increase in excitability/ calcium peak frequency under baseline conditions (Figure 2D, major point 3). This possibility should be discussed in addition to previous work indicating reduced glutamate decarboxylase function in PLP KOs. Second, could the lack of a decrease in mIPSC amplitude after TTX in TKO neurons also oppose increased network excitability? This possibility should be also discussed.

5. TTX treatment results in a rapid regulation of Hlf and Tef mRNA on the hour time scale (Figure 1C). The mEPSC/IPSC recordings were carried out in the presence of TTX. It is thus crucial to mention the time window during which WT recordings were conducted with regard to the beginning of TTX incubation to assess whether Hlf or Tef transcripts could be increased in the control group.

6. I suggest using the SI unit Hz instead of peaks/minute for the calcium imaging data.

7. The figure labels "mutant" could be more specific (e.g., Hlf,Dbp,Tef TKO).

8. Some figures are not referenced in the text.

9. It should be briefly described how miniature PSCs and APs were analyzed (e.g. template matching, threshold, or the likes…).

*Reviewer #3 (Recommendations for the authors):*

I have a few suggestions for the authors that may strengthen the manuscript:

1. Quantification of the network activity homeostasis (Figure 2 and its supplemental figure): The authors provided quantification of frequency of Ca events (probably associated with spike bursts). It is unclear why the amplitude was not taken into account. Mean firing rate should be better reflected by frequency x amplitude calculation.

2. What is the mechanism that links spike silencing to activation of PAR-bZIP TFs by inactivity? Is it mediated by the previously described mechanisms related to T-type Ca channels (Schaukowitch et al. Cell Reports 2017) or L-type Ca channels and CaMK4 (Li et al. Cell 2020)? This should be at least discussed.

3. I am not sure I agree with the interpretation of mIPSC results. The results show that both frequency and amplitude homeostatic down-regulation are disabled by TF deletion. This, in my view, may be a possible compensation to prevent a further increase in the E/I ratio (meaning that without it epilepsy would be even more severe).

Statistics: I could not find cell numbers (n) in the legends / text for each experiment.

line 179 – did the authors mean removal of silencing?

---

## [Author Response]

Essential revisions:All three reviewers acknowledge that the study provides novel insights into a long-standing question about the mechanisms that constrain homeostatic synaptic plasticity, and the findings are exciting and of broad interest. However, a couple of major concerns have been raised regarding (a) the interpretation of calcium imaging data and (b) data quantification, effect sizes and statistics in a number of experiments.

Please see detailed replies below.

(a) The main conclusion that PAR bZIP transcription factors constrain the network activity via regulating excitatory synapses should be further strengthened by addressing the following two points:(i) Ruling out the contribution of changes in inhibitory synaptic transmission in TKO neurons. If the authors cannot rule out the contribution, then calcium imaging experiments should be conducted also in excitatory or inhibitory neurons to clarify the relative contributions.

Previously we did some pilot experiments trying to attribute individual cellular firing to pyramids or subtypes of interneurons. We do not think this will be informative in the TTX condition (in either genotype) because the network activity is so intense and is shared across almost all cells in the network. Both interneurons and pyramids receive both excitation and inhibition, but the recurrent excitation is so powerful that all cells fire near simultaneously. Hence we do not believe we can clarify their relative contributions in this way. Therefore, we think that the more difficult experiment of manipulating HLF and TEF independently in pyramids and interneurons, and/or sparsely in individual cells will be much more informative, but this will take significant additional work.

(ii) In addition, calcium imaging, as the key physiology data, should be performed in WT and TKO cultures for two time points: after 2 days in TTX and after 5 days in TTX. Monitoring the kinetics will help in better relating the physiology data to the RNA-seq data and to clarify whether TKO cultures overshoots or if they show a faster compensation.

We performed additional experiments after 5 days in TTX as well as additional experiments to expand the sample size at 2 days in TTX. We report the data in Figure 2—Figure Supplement 1C-D. Response to TTX in 5 day-treated cultures is comparable to that at 2 days in TTX and the TKO phenotype is similar with the longer TTX incubation.

(b) The authors should fully address the concerns regarding data quantification and statistics that are raised by Reviewers 2 and 3.

We address these concerns and other concerns raised by each reviewer below.

Reviewer #1 (Recommendations for the authors):1. The initial RNA seq analysis in Figure 1A,B compares control vs. TTX in L5-6 pyramidal and PV+ neurons. How well does the observation of strong upregulation of HFL and TEF extend to pyramidal and PV+ cells for a 2-day TTX treatment that is used for much of the later experiments? Notably, calcium imaging based on synapsin promoter-driven GCaMP6f expression does not specify cortical layers being targeted nor discriminate between excitatory and inhibitory neurons.

Data showing upregulation of *Hlf* and *Tef* at multiple time points following initiation of TTX treatment is shown in Figure 1C. Real-time PCR experiments from RNA extracted from whole cortical slice cultures were performed at 4 hours, 1 day, 3 days and 5 days. Due to the small number of replicates, the data were underpowered for full posthoc testing of all genes/time points. However, t-tests revealed significant upregulation of both *Hlf* and *Tef* relative to Rpl10 after 24 hours in TTX (p = 0.018, 0.012).

2. Figure 2. It would be informative to monitor calcium dynamics in a cell-type specific manner to decipher the differences in homeostatic response between WT and mutant neurons.

We chose not to monitor calcium dynamics in a cell-type specific manner because, following TTX treatment, the activity of nearly all neurons in the slice are highly correlated in both genotypes (synchrony near 0.95) and are driven by strong recurrent connectivity. Hence, we do not believe there would be informative differences. We do believe that selectively manipulating *Hlf* and *Tef* in excitatory vs. inhibitory neurons would be instructive, but unfortunately the knockout available to us is a germline deletion. We plan to address this issue with additional experiments in the future, but this is likely to require very significant additional work and so is outside of the scope of this study.

3. Line 266-269. It seems premature to conclude that homeostatic plasticity of mIPSCs is not driving the TTX-dependent change in network activity based on the fact that the levels of mIPSC amplitude and frequency are indistinguishable between WT and mutant after TTX treatment. In particular, the striking reduction in mIPSC amplitude in the baseline in the mutant prep indicates that already some form of compensation for altered mIPSCs has taken place to keep the network activity level comparable between WT and mutant prior to TTX treatment. In addition, the observed lack of a further change in mIPSCs could be due to saturation of mIPSC-dependent mechanisms.

We acknowledge that the reviewer is correct and emended our statement:

“Thus, both the amplitude and frequency of mIPSCs received by pyramidal neurons after activity deprivation are indistinguishable from the wild-type in the TKO slices, suggesting that homeostatic plasticity of mIPSCs is not *directly* driving the changes in the network activity following TTX treatment. We cannot rule out the possibility that although baseline activity is normal, the baseline reduction in mIPSC amplitude contributes to subsequent network plasticity indirectly by contributing to induction of the changes in EPSCs observed, or by inducing some form of compensation other than the physiological parameters measured.”

Reviewer #2 (Recommendations for the authors):I have concerns, mainly revolving around statistics and effect sizes:1. Based on the RNA seq data shown in figure 1, it is unclear why the authors focused their analyses on Hlf, Tef and Dbp, because the data suggest more pronounced regulation of other genes. The authors should explain why these candidates were chosen for further analysis in the Results section.

We added additional explanation to this section of the paper as follows:

“To identify potential transcriptional regulators of homeostatic plasticity, we looked for transcription factors (TFs) upregulated in the TTX condition. TFs can be classified into families, and in some cases subfamilies, which share a DNA binding domain, and so are expected to bind the same target DNA sequences in the genome and therefore to regulate the same or similar target transcripts. We asked which families or subfamilies were most over-represented amongst the TFs differentially expressed during activity blockade. The most overrepresented subfamily was the PAR bZIP subfamily of TFs, a subtype of the CEBP-related family in the class of Basic leucine zipper factors (classification data from tfclass.bioinf.med.uni-goettingen.de Wingender et al. 2015 ). This subfamily includes 3 transcriptional activators Hlf, Tef, and Dbp, and a transcriptional repressor, Nfil3.”

2. One of the central findings of this study is that Hlf-/-/Dbp-/-/Tef-/- triple knockout (TKO) slice cultures display a more pronounced increase in calcium peak frequency upon TTX treatment compared to WT controls. In the Results section, it is mentioned that TTX treatment induces a two-fold increase in calcium peak frequency in WT, and a four-fold increase in TKO cultures (l. 142, 143). However, based on the mean data plotted in figure 2D, the increase in WT is around 3-4-fold (15 peaks x s^-1/4 peaks x s^-1=3.5), whereas the one in TKO is around four-fold (30/7.5=4). Could the authors comment on the discrepancy between the text and the data? Furthermore, it is key to test for an interaction between treatment and genotype to support this major conclusion by statistics. Additionally, it would be helpful to plot the relative increase in calcium peak frequency upon TTX treatment in WT and TKO.

We thank the reviewer for catching the discrepancy. We performed additional experiments and have re-analyzed the old and new data together. We have fixed the text to reflect the data and have plotted the peak frequency normalized to control (Supplemental Figure 2A) as well as non-normalized frequency in Hz. The combined data and analysis further strengthen our conclusion that there is no change in network activity level at baseline between WT and TKO cultures.

We tested for interactions and found highly significant (<0.0001) interaction between treatment and genotype. We include the results in the text (line 141).

“While 2 days of TTX incubation increases the peak frequency 3.6-fold in WT slices (from 0.06 +/- 0.01 to 0.22 +/- 0.04 Hz), the same silencing duration produces a nearly six-fold increase in the TKO slices (from 0.09 +/- 0.02 Hz to 0.5 +/- 0.05 Hz; Figure 2, Figure 2—Figure Supplement 1A). A two-way ANOVA revealed significant effects of treatment (TTX vs. Control; p<0.0001) and genotype (p<0.0001) and a significant interaction between the two (p<0.0001). Post hoc T-tests (with Tukey correction for multiple comparisons) revealed that TTX increases in activity were highly significant in both WT (p = 0.0007) and TKO (p<0.0001) slices and that TTX produced a significantly stronger increase in the TKO (p<0.0001), while the baseline activity of cells from slices derived from the TKO and WT do not differ significantly (p=0.92). Consistent with this, normalizing the TTX response to the control response revealed a larger relative increase in TKO than WT cultures (Figure 2—Figure supplement 1A).”

3. On a similar note, although not statistically significant, the mean calcium peak frequency in TKO cultures is around two-fold higher than WT under baseline conditions (Figure 2D). A similar trend can be seen in the data shown in figure 6A (top panel), whereas the baseline calcium peak frequency is similar between TKO and controls in the experiment shown in Figure 2B. Based on the scatter of the data, can the authors exclude that baseline calcium peak frequency is increased in TKO cultures? I suggest running a power analysis for these data to verify if the sample sizes are sufficient to support their conclusions. Otherwise, sample sizes would have to be increased, or the conclusions should be revised/toned down. Alternatively, could additional parameters, such as inter-burst interval, be read out to support the conclusions that baseline network excitability is unchanged in TKO cultures, and that TTX application leads to a more pronounced increase in calcium peak frequency? Another possibility may be to pool the data of the different experiments?

We performed additional experiments, reanalyzed our existing data and replotted the figures including the additional experiments. A small subset of experiments were excluded because the calcium signal was too low to accurately measure changes in firing. We have made the exclusion criteria clear in the methods. The previous trend towards an increase at baseline is no longer present with the additional data (Figure 2D). Additionally, we now report synchrony, a measure of the average cross-correlation at zero lag during network bursts (figure 2 supplement 1B). The synchrony metric saturates in the WT TTX condition, making it unsuitable for detecting further increases in network activity and so we use peak frequency for most of the rest of the manuscript. We also performed a power analysis. The synchrony results and power analysis are described in the following sentences (line 152):

“The TTX condition also increased synchrony (from 0.76 +/- 0.03 to 0.95 +/- 0.01) in WT, and in the TKO (from 0.74 +/- 0.03 to 0.94 +/- 0.05; Figure 2—Figure Supplement 1B). While the TTX effects were highly significant (Two-way ANOVA; p<0.0001), there was no significant effect of genotype (p=0.43) and no interaction between treatment and genotype (p=0.65), presumably reflecting the fact that synchrony is already saturated by TTX treatment in the WT and cannot further increase in the TKO, but also indicating that baseline synchrony is not altered in the TKO. A power test revealed that with the effect size and variances observed, 95 slice cultures in each group would be needed to detect a significant difference between WT and TKO peak frequency, while for the synchrony measure, the required N would be 1609.”

4. Based on the data shown in figure 6, they conclude that one copy of Tef is sufficient to constrain homeostatic plasticity. Can they rule out that one copy of Hlf is sufficient as well? This would actually fit to the result that one copy of Tef or Hlf rescues lifespan in TKO. The peak frequencies of DKO;Tef +/- and DKO;Hlf +/- after TTX treatment actually look quite similar. Are these two groups statistically different from one another? Only comparing the absolute DKO;Tef +/- or DKO;Hlf +/- data to WT is not very informative. Again, I suggest normalizing the data of each genotype to the respective control, and to test for interactions between treatment and genotype.

We agree with the reviewer’s skepticism of our ability to distinguish the contributions of Hlf and Tef and wrestled with how best to perform this analysis. A key limitation is that there are 27 potential genotypes (Null, Het, wild type) for the three genes considered jointly and so even with a total of 185 experiments (87 Control and 98 TTX) we were almost definitely underpowered for most specific comparisons. Rather than attempt to make all possible pairwise comparisons, we chose to put all of the data together (the 88 experiments shown in figure 2 and figure 2—figure supplement 1, as well as 97 additional experiments that included those shown in the earlier version of figure 6 as well as some additional experiments we performed in the past year). We then fit these data with a series of linear regression models to determine the relationship between peak frequency and genotype. The same set of six models were used to separately fit the data from control slices and from TTX-treated slices.

These analyses are described in the following section (line 258):

“To test whether Hlf, Tef, and Dbp work together to regulate homeostatic plasticity or if they have unequal contributions to restraining network response to activity deprivation, we measured the homeostatic response while varying gene copy number of each transcription factor. In addition to the WT and TKO data described in Figure 2 and Figure 2—Figure Supplement 1, we measured calcium transients in slice cultures treated with or without TTX from 97 additional animals with various combinations of 0, 1 or 2 alleles of Hlf, Tef and Dbp (full details of each experiment given in associated data file). Rather than attempt to statistically test individual differences between each of 27 potential genotypes (Null, Het, wild type for the three genes considered jointly), we fit a series of multivariate linear models to test the ability of genotype to predict calcium peak frequency. The same six models, described in Table 1, were fit separately to data from control and TTX-treated slices. The models, shown in Table 1, differ in how they depend on the number of alleles of each of the three genes. Model 1 depends on all three genes, models 2-4 depend only on one of the three genes (each in turn) and models 5 and 6 depend only on Hlf and Tef, either independently, or summed together. The ability of each model to account for the data was estimated from the adjusted R^2^ which reflects the fraction of the variance in the data predicted by the model variables. Several key points were evident from this analysis:

1) All of the models produced poor fits (adjusted R^2^ of -0.01 to 0.063) to the control data, consistent with the hypothesis that baseline activity does not depend on genotype.

2) All of the models that included variables for Hlf and/or Tef produced much better fits to the TTX data (adjusted R^2^ of 0.198 to 0.296) consistent with the hypothesis that rebound activity is influenced by genotype. In these models, the coefficients (effect size) for Hlf and Tef were similar and T-tests revealed they were highly significantly different from zero.

3) The model that only depended on Dbp (line 4) provided a poor fit to the TTX data (adjusted R^2^ = 0.073) and in the model that included both terms for Dbp and the other two factors (line 1), the Dbp coefficient was small and not significantly different from zero. These observations are consistent with the hypothesis that rebound activity does not depend on Dbp and that even in a significant model including Hlf and Tef, Dbp does not add additional explanatory power to the model.

3) The models that included terms for both Hlf and Tef (line 5) or for their sum (line 6) produced the best fits (adjusted R^2^ = 0.291 and 0.296); i.e. better than those that depend on only one of these factors (lines 2,3; adjusted R^2^ = 0.198, 0.229). This is consistent with the hypothesis that both genes contribute to constraining cortical homeostatic plasticity.

Since the model with the highest adjusted R^2^ was that which depended only on the sum of the number of Hlf and Tef alleles (line 6), we used this model for further post hoc (Tukey) tests to determine the effect of different numbers of alleles on the TTX response. The results of this analysis are shown in Figure 6. The most significant differences were between TKO slices (0 alleles; n=32, mean=0.48) and the other genotypes tested (1 allele, n=16, mean=0.32; 2 alleles, n=26, mean=0.35; and 4 alleles, n=24, mean=0.23; p <0.0001 to 0.0014). Most of the other comparisons were not significant, with the exception of the difference between WT responses and those for animals with only two alleles (p=0.011). This indicates that most of the difference between TKO (zero alleles of Hlf + Tef) and WT (4 alleles of Hlf + Tef) could be restored by a single allele, although there was a small but significant improvement between 2 and 4 alleles. The fact that this difference was significant, but the differences between 1 and 2 alleles, and 1 and 4 alleles were not, may reflect a real difference between these groups, or may reflect the imbalanced numbers of animals in each group. The group with 2 alleles included animals that were WT for Hlf but lacked Tef (N=8), animals that were WT for Tef and lacked Hlf (N=9), and trans-Het animals that were heterozygous for both Hlf and Tef (N=7). A separate analysis of variance of these three subgroups revealed that the means (0.32, 0.41 and 0.33) were not significantly different (p=0.28), consistent with the hypothesis that alleles of Hlf and Tef can substitute for one another in their ability to regulate the homeostatic response to activity deprivation.”

5. Sample sizes (culture number/ if applicable, neuron number) should be clearly specified for each experiment. Are some data sets only based on two cultures (e.g. Figure 6 ctrl.)? Again, a power analysis may be helpful to determine sample sizes.

We performed additional experiments and added additional data to figure 6. The N’s for each major group are stated in the legend (#cultures) and the precise genotypes for every experiment and the # neurons selected in each slice are provided in the accompanying supporting data file. We did not perform a power analysis for each model.

6. The major (recorded) effect underlying the increase in network excitability is an increase in mEPSC frequency. Does this indicate that the mechanisms that constrain homeostatic plasticity mainly work on presynaptic mechanisms, or an increase in synapse number? These possibilities should be also discussed.

We add a discussion of potential mechanism underlying this change to the Discussion section of the manuscript:

“Although two days of activity blockade in dissociated cortical cultures were initially found to produce scaling up of EPSC amplitudes without accompanying changes in mEPSC frequency (Turrigiano et al., 1998), multiple subsequent studies have found evidence for changes in both amplitude and frequency as found here (Wierenga et al., 2006; Koch et al., 2010; Echegoyan et al., 2007; Thiagarajan et al., 2005). Changes in mini frequency are typically interpreted as reflecting increased presynaptic release, either via increases in the numbers of synapses or active release sites, or via changes in the rate at which spontaneous, action potential independent release occurs at each site.”

7. TTX application increases mEPSC frequency by around two-fold and does not affect other recorded parameters, such as excitability. How could a two-fold increase in mEPSC frequency translate into a four-fold increase in calcium peak frequency? Is there any evidence for a supralinear increase in network activity with regard to miniature frequency? This issue should be discussed.

With inclusion of additional data, the increase in mEPSC frequency following TTX was 1.7-fold larger in the TKO than in WT (Figure 3B, while the change in network activity measured from calcium peak frequency was 1.6-fold greater Figure 2D) at 2 days and 2.5-fold greater at 5 days. We added discussion of the relationship in the following paragraph:

“Although the increase observed in mEPSC frequency (1.7-fold greater in TKO than WT), was lower than the relative increase in activity levels, measured from the frequency of calcium transients (2.5-fold greater in TKO than WT at 5 days), these two measures need not be linearly related. First, mini frequency can be distinct from action potential evoked release (Crawford et al., 2017) and second, in a highly recurrent network, the relationship between firing and synaptic strength can be nonlinear (Toyoizumi and Abbott, 2011).”

8. What is the evidence for 'exaggerated' homeostatic plasticity in TKO cultures? Can they rule out that TKO neurons compensate better? This possibility should be discussed unless they can exclude the possibility that TKO cultures indeed do not retarget set-point activity.

Although we are not positive we understand the point the reviewer is making, we think he/she is suggesting that we do not know whether the genetic manipulation changes rebound activity following activity deprivation because of a change in the vigor or speed with which the network attempts to restore activity (which we refer to as exaggerated homeostatic plasticity) or a change in the set point to which activity is trying to return, or both.

We directly tested this possibility in the experiment shown in figure 2 supplement 2. We observe that in the TKO, rebound activity is indeed higher, but it returns to the same baseline level after two days of recovery in normal media lacking TTX. We realize that we did not adequately highlight the significance of this result and thank the reviewer for spurring us to make this clearer. We added the following sentences to the Results section of the manuscript (line 174):

“In addition, changes in activity following rebound from deprivation could reflect either a change in the vigor of the circuit’s attempt to restore activity (i.e. enhanced homeostatic plasticity) or a persistent change in the setpoint to which activity is returned (Styr et al., 2019), or both. To test whether the transcriptional restraint of homeostasis is required for bidirectional flexibility and to assess whether the setpoint had changed, we asked whether the network is able to return back to baseline activity levels when the activity blockade is removed. We measured network activity immediately after the end of silencing with TTX as well as following two days of recovery in TTX-free media. In WT slices, activity deprivation causes hyperactivity, evident from a two-fold rise in the peak frequency of the calcium activity. However, this exuberant activity is restored back to baseline levels within two days following restoration of action potential firing. In the mutant, even though the initial response to TTX is exaggerated, the network is also able to return to baseline levels following two days of recovery, to levels indistinguishable from WT (Figure 2—Figure Supplement 2). These data suggest that the recovery from a high activity state is not diminished in the TKO and suggests that the mutation in fact produces exaggerated homeostatic plasticity rather than a change in activity setpoint.”

Other comments:1. It would be very helpful if the abstract could contain more specific information (e.g. experimental system, what kind of homeostatic plasticity was studied, some specific results etc.)

We have reworked the abstract to be more descriptive and to include these details.

2. The more pronounced increase in mEPSC frequency upon TTX treatment in TKO cultures compared to WT (Figure 3B) appears more robust than the corresponding changes in calcium peak frequency (Figure 2). However, they should also test for an interaction between treatment and genotype to support their conclusion by statistics for the data shown in figure 3B, C.

We now include these statistical analyses including the interaction between treatment and genotype in the figure legend.

3. Similarly, although not statistically significant, the decrease in mIPSC frequency upon TTX treatment seems less pronounced in TKO cultures (Figure 4B, C). Could they run a power analysis and test for treatment-group interaction as well?

We now test for interactions between the treatment and the genotype and report the findings in the figure legend along with a power analysis of the sample size that would be required for the observed difference to be significant:

“Two-way ANOVA revealed a significant main effect of TTX treatment (p<0.0001) but not genotype (p=0.54). There was no significant interaction between treatment and genotype (p=0.40), N=28 for WT control, N=37 for WT TTX, N=23 for TKO control, N=20 for TKO TTX. TTX treatment decreased mIPSC frequency in WT (p=0.0007, post hoc Tukey test) but not in TKO cells (p=0.14, post hoc Tukey test). mIPSC frequency in TTX-treated cells is not different in TKO slices compared to WT (p=0.72). The normalized change in frequency (TTX normalized to Control) was 0.55 in WT and 0.70 in TKO. We performed a power analysis that revealed that for the observed effect size and variance a sample of 68 neurons per condition would be required for significance.”

4. mIPSC amplitudes are decreased in TKO cultures, and there is no further decrease upon TTX application (Figure 4D, E). First, the decrease in baseline mIPSC amplitude in TKO is interesting, and may represent a homeostatic response that counteracts a potential increase in excitability/ calcium peak frequency under baseline conditions (Figure 2D, major point 3). This possibility should be discussed in addition to previous work indicating reduced glutamate decarboxylase function in PLP KOs. Second, could the lack of a decrease in mIPSC amplitude after TTX in TKO neurons also oppose increased network excitability? This possibility should be also discussed.

The decrease in mIPSC amplitude observed would not be expected to represent a homeostatic response counteracting increased excitability/ calcium peak frequency since it would exacerbate the resulting increase in network excitability. Furthermore, as noted above, additional experiments make it clearer that baseline frequency is not altered. The prior work on reduced glutamate decarboxylase function is discussed in the following paragraph:

*“*Gachon et al. (Gachon et al., 2004) first reported that deletion of the PARbZIP proteins TEF, HLF and DBP caused spontaneous and audiogenic seizures. They suggested that this may reflect effects on neurotransmitter metabolism caused by a reduction in pyridoxal phosphate (PLP) since pyridoxal kinase, which catalyzes the last step in the conversion of vitamin B6 into PLP is a target of the PAR bZIP proteins. PLP is a required cofactor for decarboxylases and other enzymes involved in the synthesis and metabolism of GABA, glutamate and the biogenic amine neurotransmitters including dopamine, serotonin and histamine. Using HPLC they found no changes in GABA or glutamate, but reduced levels of dopamine, serotonin and histamine in the knockout animals. Although changes in amine transmitters could potentially contribute to phenotypes in vivo, they are unlikely to contribute to plasticity in cortical slice cultures which do not include the brainstem nuclei from which these projections arise. Reduced glutamate decarboxylase action could potentially have contributed to the observed baseline reduction in mIPSCs since heterozygous (Lazarus et al., 2015) and homozygous (Lau and Murthy, 2012) mutants of Gad1, the gene encoding the GAD67 isoform of glutamic acid decarboxylase, have reduced mIPSC amplitudes. However, this does not account for the enhanced homeostatic overshoot following activity blockade, since in the TKO animals, activity blockade did not produce any further reduction in inhibition but produced robust changes at excitatory synapses.”

5. TTX treatment results in a rapid regulation of Hlf and Tef mRNA on the hour time scale (Figure 1C). The mEPSC/IPSC recordings were carried out in the presence of TTX. It is thus crucial to mention the time window during which WT recordings were conducted with regard to the beginning of TTX incubation to assess whether Hlf or Tef transcripts could be increased in the control group.

We added the following detail to the methods section of the manuscript.

“The slices were incubated in TTX only during active recording, which lasted no more than 30 minutes for each slice.”

6. I suggest using the SI unit Hz instead of peaks/minute for the calcium imaging data.

We re-plotted all of our data in Hz.

7. The figure labels "mutant" could be more specific (e.g., Hlf,Dbp,Tef TKO).

We relabeled our figures as suggested.

8. Some figures are not referenced in the text.

We have now included reference to all the figures in the text.

9. It should be briefly described how miniature PSCs and APs were analyzed (e.g. template matching, threshold, or the likes…).

We added the following description to the methods*:*

“Threshold-based detection of mPSCs and action potentials were implemented using custom-written IGOR scripts equipped with routines for baseline subtraction, custom filtering and measurements of intervals, amplitudes and kinetic properties.”

Reviewer #3 (Recommendations for the authors):I have a few suggestions for the authors that may strengthen the manuscript:1. Quantification of the network activity homeostasis (Figure 2 and its supplemental figure): The authors provided quantification of frequency of Ca events (probably associated with spike bursts). It is unclear why the amplitude was not taken into account. Mean firing rate should be better reflected by frequency x amplitude calculation.

Although in principle the amplitude of calcium transients may provide additional information about the magnitude of the firing underlying these events, we found that the distribution of event amplitudes were dependent upon other experimental factors such as the amount of calcium indicator expressed as well as optical factors such as the depth of the cell in the slice. Although it may be possible with further calibration experiments to control for some of these factors, we decided instead to simply use the calcium transients as indicative of periods of neuronal firing without attempting to ascertain the precise number of action potentials contributing. We added the following summary of this idea to the methods:

“Since the distribution of event amplitudes across cells was potentially dependent on other experimental factors such as the amount of calcium indicator expressed as well as optical factors such as the depth of the cell in the slice, we decided not to try to infer the magnitude of firing from the amplitude of events, but only periods of elevated firing from the timing of events.”

2. What is the mechanism that links spike silencing to activation of PAR-bZIP TFs by inactivity? Is it mediated by the previously described mechanisms related to T-type Ca channels (Schaukowitch et al. Cell Reports 2017) or L-type Ca channels and CaMK4 (Li et al. Cell 2020)? This should be at least discussed.

We added the following speculative paragraph to the discussion:

“The mechanisms linking activity deprivation to Tef and Hlf activation were not directly addressed in this work, but seem likely to depend on reduced calcium influx. Prior studies have suggested alternately that scaling up of mEPSC amplitudes depends upon calcium influx through either T-type (Schaukowitch et al., 2017) or L-type (Li et al., 2020) calcium channels. The former pathway has also been found to depend on the activity-dependent transcription factors Elk1 and SRF. Activation of the constraining TFs Tef and Hlf could be secondary or delayed responses dependent on changes in the transcription and translation of other activity-dependent TFs, or may represent a primary response to activity deprivation, analogous to, but in the opposite direction of the well-studied immediate early genes such as Fos, Arc and others.”

3. I am not sure I agree with the interpretation of mIPSC results. The results show that both frequency and amplitude homeostatic down-regulation are disabled by TF deletion. This, in my view, may be a possible compensation to prevent a further increase in the E/I ratio (meaning that without it epilepsy would be even more severe).

We agree that we can’t exclude this possibility and add the following statement to the discussion:

“We cannot rule out the possibility that although baseline activity is normal, the baseline reduction in mIPSC amplitude contributes to subsequent network plasticity indirectly by contributing to induction of the changes in EPSCs observed, or by inducing some form of compensation other than the physiological parameters measured”

Statistics: I could not find cell numbers (n) in the legends / text for each experiment.

We thank the reviewer for catching the error and now report N values for each experiment.

line 179 – did the authors mean removal of silencing?

We have corrected the wording and thank the reviewer for pointing it out. The statement now reads:

“We measured network activity immediately after the end of silencing with TTX”ab